# Isoginkgetin derivative IP2 enhances the adaptive immune response against tumor antigens

Romain Darrigrand[1,8], Alison Pierson[1,8], Marine Rouillon[1,2], Dolor Renko[3], Mathilde Boulpicante[1], David Bouyssié[4], Emmanuelle Mouton-Barbosa[4], Julien Marcoux[4], Camille Garcia[5,7], Michael Ghosh [6], Mouad Alami[3] & Sébastien Apcher [1✉]

The success of cancer immunotherapy relies on the induction of an immunoprotective response targeting tumor antigens (TAs) presented on MHC-I molecules. We demonstrated that the splicing inhibitor isoginkgetin and its water-soluble and non-toxic derivative IP2 act at the production stage of the pioneer translation products (PTPs). We showed that IP2 increases PTP-derived antigen presentation in cancer cells in vitro and impairs tumor growth in vivo. IP2 action is long-lasting and dependent on the CD8$^+$ T cell response against TAs. We observed that the antigen repertoire displayed on MHC-I molecules at the surface of MCA205 fibrosarcoma is modified upon treatment with IP2. In particular, IP2 enhances the presentation of an exon-derived epitope from the tumor suppressor nischarin. The combination of IP2 with a peptide vaccine targeting the nischarin-derived epitope showed a synergistic antitumor effect in vivo. These findings identify the spliceosome as a druggable target for the development of epitope-based immunotherapies.

[1] Université Paris-Saclay, Institut Gustave Roussy, Inserm, Immunologie des tumeurs et Immunothérapie, Villejuif, France. [2] SATT Paris Saclay, Orsay, France. [3] Université Paris-Saclay, CNRS, BioCIS, Châtenay-Malabry, France. [4] Institut de Pharmacologie et de Biologie Structurale (IPBS), Université de Toulouse, CNRS, UPS, Toulouse, France. [5] Institut Jacques Monod, CNRS U7592 Université Paris Diderot, Paris, France. [6] Department of Immunology, Interfaculty Institute for Cell Biology, University of Tübingen, Tübingen, Germany. [7] Present address: Institut Pasteur, Unité de Spectrométrie de Masse pour la Biologie (MSBio), Centre de Ressources et Recherches Technologiques (C2RT), USR 2000 CNRS, Paris, France. [8] These authors contributed equally: Romain Darrigrand, Alison Pierson. ✉email: sebastien.apcher@gustaveroussy.fr

All nucleated cells of jawed vertebrates present antigenic peptides (APs) at their surface through the major histocompatibility complex class I (MHC-I) presentation pathway. MHC-I peptides (MIPs) are 8- to 12-amino acid-long and reflect the inherent metabolic cellular activity[1]. Initial clinical trials investigating vaccines targeting tumor antigens (TA) have not met expectations. The main failures have been associated with immunosuppressive mechanisms and with a suboptimal choice of antigens[2,3]. One of the important events that drives tumor escape from immunosurveillance and is correlated with a poor prognosis is the loss or the downregulation of MHC-I antigen presentation by tumor cells[4,5]. The latter take advantage of defects in components of the MHC-I presentation pathway to escape cytotoxic T lymphocyte (CTL) and natural killer (NK) cell recognition[6]. Moreover, along with the overall decrease in MHC-I antigen presentation, the very nature and the number of antigens presented at the cell surface, namely the MHC-I immunopeptidome, is of critical importance for immune recognition. For example, the loss of expression at the tumor cell surface of a specific TA identified and targeted with immunotherapy, such as Her/neu in breast cancer or CEA in colon cancer, can lead to immune evasion[7,8]. To counter this phenomenon, current strategies aim at enlarging the range of targetable tumor antigens and restoring MHC-I antigen presentation[9–12].

In order to understand the dynamics of the MHC-I immunopeptidome, we focused on the source of APs for the MHC-I presentation pathway. Exploring the concept of defective ribosomal products (DRIPs)[13–16], we showed that one of the most important sources of APs is a pioneer translation event that occurs on pre-mRNAs, before introns are spliced out, and independently of the translation of the corresponding full length proteins. The pioneer translation products (PTPs) produced through this non-canonical translation event can therefore derive from intronic sequences, 3' or 5' untranslated regions (UTR) as well as alternative reading frames[17,18]. The discovery of PTPs emphasizes the existence of a nuclear translation mechanism of precursor mRNAs that participates to the production of MHC-I peptides (MIPs). Moreover, PTPs play a role in the dynamic of cancer development. When inoculated into mice, it has been shown that cancer cells presenting PTP-derived antigens at their surface can be recognized by specific T cells leading to tumor growth reduction[19]. Besides, purified PTPs encompassing a model immunodominant epitope efficiently promote anticancer immune response when injected as a peptide vaccine into mice.

Cancer therapies have been constantly evolving with the hope of finding the most effective agents to eradicate tumors with the least toxic effects on healthy tissues. Unlike conventional cancer treatments that focus on tumor cells, immunotherapies target the immune system to strengthen and stimulate the patient's own defenses against cancer. Breakthrough of the last decade, the immune checkpoint inhibitors against cytotoxic T lymphocyte associated protein 4 (CTLA-4), programmed death 1 (PD-1) and programmed death ligand 1 (PD-L1) have shown unprecedented antitumor effect in particular against advanced melanoma or non-small cell lung cancer[20,21]. However, the efficiency of those monoclonal antibodies is limited to a fraction of patients and is often associated with deleterious side effects[22]. Attention is being paid to small molecules with immunomodulatory properties, which offer a cost-effective production, a reversible action and reduced systemic side effects due to their short half-lives[23].

The biflavonoid isoginkgetin is a natural compound extracted from the leaves of the *Ginkgo biloba* tree. Multiple properties of the molecule have been reported and associated with its antitumor activity. Isoginkgetin was first shown to inhibit tumor cell invasion by inhibiting the production of the matrix metalloproteinase 9 (MMP-9)[24]. Indeed, isoginkgetin-induced downregulation of the

NF-κB pathway leads to the upregulation of the MMP-9 inhibitor (TIMP-1) in human fibrosarcoma. More recently, it has been demonstrated that isoginkgetin inhibits 20 S proteasome activity and induces a toxic accumulation of polyubiquitinated proteins[25]. Eventually, isoginkgetin was described as a general inhibitor of pre-mRNA splicing, which stalls spliceosome assembly at the prespliceosomal A complex[26]. Pre-mRNA splicing is catalyzed in the nucleus by the spliceosome, a conserved and dynamic multi-RNA/protein complex composed of five small nuclear RNAs (snRNAs) in interaction with over 180 proteins[27]. A growing number of studies report that the deregulation of the spliceosome complex entails aberrant splicing patterns in many cancers contributing to abnormal tumor cell proliferation and progression[28–31]. In a recent study, we observed that splicing inhibition positively modulates the presentation of a PTP-derived model antigen in HEK-293T cells treated with isoginkgetin[18].

Here we show that the biflavonoid isoginkgetin and its water-soluble derivative IP2 enhance the presentation of PTP-derived antigens at the surface of cancer cells in vitro. In addition, IP2 induces a long-lasting anticancer immune response in vivo. Finally, IP2 was shown to reshape the MHC-I immunopeptidome of MCA205 fibrosarcoma. Our findings shed light on a new immunomodulatory agent whose antitumor activity relies on the induction of immunogenic epitopes that can be targeted in the context of epitope-based immunotherapies.

## Results

**Isoginkgetin increases exon- and intron-derived SL8 presentation in cancer cells in vitro and inhibits the growth of SL8-expressing tumors in vivo in an immune-dependent manner.** In order to improve the antigenicity of cancer cells and thus their recognition by the immune system, we determined whether isoginkgetin was able to enhance the expression and the presentation of tumor-associated PTP-derived antigens. For that purpose, the murine MCA205 fibrosarcoma and B16F10 melanoma transiently expressing the intron-derived SL8 epitope within the β-Globin gene construct (globin-SL8-intron) were treated with increasing doses of isoginkgetin up to the limit of IC$_{50}$ determined by MTT assay (Supplementary Fig. S1a). In accordance with our previous study, isoginkgetin elicited an increase in the intron-derived SL8 antigen presentation, in a dose dependent manner (Fig. 1a). To further investigate the impact of isoginkgetin on PTP presentation, MCA205 and B16F10 cell lines transiently expressing the exon-derived SL8 epitope within the β-Globin gene construct (globin-SL8-exon) or the splicing-independent OVA cDNA (OVA-derived SL8) were treated with increasing doses of the compound. We observed that isoginkgetin increases splicing-dependent but not splicing-independent SL8 presentation in a dose dependent manner (Fig. 1b, c). Furthermore, we observed that the expression of the MHC-I H-2K$^b$ molecules at the cell surface is differently affected upon treatment with isoginkgetin depending on the cell type (Supplementary Fig. S1b). Those variations are therefore not correlated with the effect of the compound on the SL8 antigen presentation in vitro. Overall, these results show that the natural product isoginkgetin acts as an enhancer of the PTP-derived antigen presentation in cancer cells independently of the epitope setting (i.e., in exonic or in intronic sequences) or the cell type. Moreover, they support the idea that pre-mRNAs are a source for antigen presentation when the splicing machinery is impaired. This suggests an action of isoginkgetin during the production stage of PTPs and not downstream in the MHC-I antigen presentation pathway.

Antigen abundance at the cell surface has been demonstrated to be a key parameter in determining the magnitude of the CD8$^+$ T-cell response[32]. Therefore, we assessed the effect of isoginkgetin

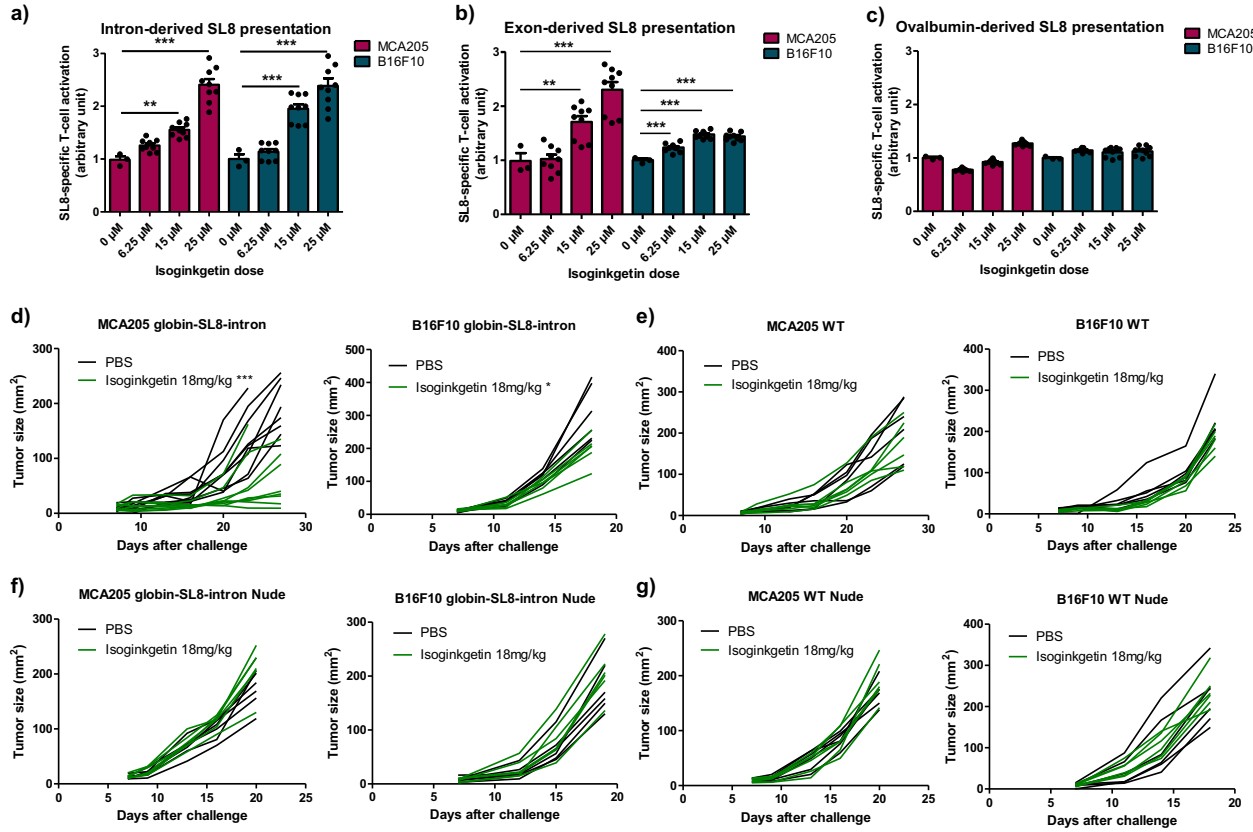

**Fig. 1 Isoginkgetin increases exon- and intron-derived SL8 presentation in cancer cells in vitro and inhibits the growth of SL8-expressing tumors in vivo in an immune-dependent manner.** SL8-specific B3Z T-cell activation after co-culture with murine MCA205 fibrosarcoma and B16F10 melanoma both transiently expressing (**a**) the intron-derived SL8 epitope, (**b**) the exon-derived SL8 epitope, or (**c**) the OVA cDNA construct and treated for 18 h with isoginkgetin at indicated doses. Unspecific B3Z T-cell activation was considered to normalize the results. Data are means ± SEM of relative B3Z T-cell activation for at least three biological replicates. **d** Individual tumor growth kinetics of MCA205 fibrosarcoma (left panel) and B16F10 melanoma (right panel) both stably expressing the globin-SL8-intron construct and (**e**) MCA205 fibrosarcoma WT (left panel) and B16F10 melanoma WT (right panel) subcutaneously inoculated into the right flank of immunocompetent C57BL/6 mice thereafter treated intraperitoneally with 18 mg/kg of isoginkgetin at days 5, 10, and 15. **f** Individual tumor growth kinetics of MCA205 fibrosarcoma (left panel) and B16F10 melanoma (right panel) both stably expressing the globin-SL8-intron construct and (**g**) MCA205 fibrosarcoma WT (left panel) and B16F10 melanoma WT (right panel) subcutaneously inoculated into the right flank of immunodeficient nude mice thereafter treated intraperitoneally with 18 mg/kg of isoginkgetin at days 5, 10, and 15. **P < 0.01, ***P < 0.001 (One-way ANOVA (**a**–**c**) and Student t-test (**d**–**g**)).

treatment on the antitumor response against SL8-expressing tumors in vivo. MCA205 fibrosarcoma and B16F10 melanoma stably expressing the globin-SL8-intron construct were inoculated subcutaneously into C57BL/6 mice. At days 5, 10, and 15 post-tumor inoculation, the mice were injected intraperitoneally with 18 mg/kg of isoginkgetin and the tumor growth was monitored. We observed a 60% decrease in the growth of MCA205-globin-SL8-intron at day 27 (Fig. 1d and Supplementary Fig. 1c, left panels) and a 30% decrease in the growth of B16F10-globin-SL8-intron tumors at day 18 (Fig. 1d and Supplementary Fig. 1c, right panels) in isoginkgetin-treated mice. To assess the link between SL8 overexpression and tumor growth reduction in vivo after isoginkgetin treatment, we performed the same experiment in mice inoculated with either MCA205 or B16F10 wild-type (WT) cells. No significant reduction in the growth of MCA205 WT (Fig. 1e and Supplementary Fig. S1d, left panels) and B16F10 WT (Fig. 1e and Supplementary Fig. S1d, right panels) was observed after treatment with 18 mg/kg of isoginkgetin. Taken together, these results show that the isoginkgetin-induced increase in PTP presentation and tumor growth reduction are mediated by the effect of the compound on the splicing of the immunodominant SL8 epitope.

We then assessed the requirement of the immune system for isoginkgetin to reduce tumor growth. Immunodeficient Nu/Nu nude mice were injected subcutaneously with either MCA205 or B16F10 cells stably expressing the globin-SL8-intron construct or WT cells and thereafter treated under the conditions previously stated. No effect of isoginkgetin treatment was observed on the growth of each of the four tumor types in nude mice (Fig. 1f, 1g and Supplementary Fig. S1e and f).

Eventually, these results show that isoginkgetin enhances the presentation of both exon-derived and intron-derived antigens in cancer cells. They also demonstrate that isoginkgetin action relies on splicing events affecting immunodominant epitopes and is dependent on the adaptive immune system.

**The derivative IP2 retains the properties of the parent isoginkgetin on antigen presentation with reduced cytotoxicity.** Two intrinsic properties of the isoginkgetin compound are limiting its further use in preclinical or clinical studies. Indeed, the molecule is cytotoxic for both normal and cancer cells and its high hydrophobicity reduces its biodisponibility. Therefore, water-soluble derivatives of the natural isoginkgetin product were tested for their ability to increase PTP-derived antigen presentation in vitro.

The derivatives IP2 and M2P2 were synthesized from the commercial isoginkgetin compound (Supplementary Fig. S2). In short, the synthesis of IP2 was accomplished by the phosphorylation of isoginkgetin employing in situ formation of diethylchlorophosphite to provide compound 1. Further cleavage of the ethyl ester protective groups with iodotrimethylsilane afforded the phosphoric acid intermediate, which was immediately treated with sodium hydroxide to complete a practical route to the disodium phosphate prodrug. For the synthesis of the M2P2 molecule, the remaining two phenol groups of compound 1 were alkylated using methyl iodide to furnish compound 2. Treatment of the latter under similar conditions to prepare IP2 from compound 1 gave the disodium phosphate prodrug M2P2. The water solubility of IP2 (27 mg/mL) and M2P2 (29 mg/mL) was found to be greatly higher than that of the parent compound isoginkgetin (0.1 mg/mL).

IP2 and M2P2 compounds were first tested for their ability to increase the MHC-I presentation of PTP-derived antigens in vitro. For that purpose, MCA205 and B16F10 cells transiently expressing the globin-SL8-intron construct were treated with 15 μM or 35 μM of IP2 or M2P2 for 18 h. While treatment with IP2 increases the intron-derived SL8 presentation in both MCA205 and B16F10 cells, as observed with isoginkgetin, M2P2 decreases its presentation in MCA205 cells and does not impact it in B16F10 cells (Fig. 2a). In contrast, the alkylating agent cyclophosphamide has no effect on antigen presentation in both cell lines and the antimetabolite gemcitabine only acts on

MCA205 cells (Supplementary Fig. S3a). In parallel, we showed that neither the expression of H-2K$^b$ molecules at the cell surface at steady state (Fig. 2b) nor the recovery of H-2K$^b$ molecules at the cell surface after acid strip (Supplementary Fig. S3b) were affected by the treatment with IP2 or M2P2. Taken together, those results suggest a specificity of action of the isoginkgetin derivative IP2 on antigen presentation that is not shared with molecules with other activities and that does not rely on increased number of MHC-I molecules at the cell surface.

Furthermore, we observed by MTT assay that IP2 compound displays no toxicity toward MCA205 and B16F10 cells in a wide range of concentrations (Supplementary Fig. S3c). More specifically, IP2 treatment does not induce tumor cell apoptosis up to 1 mM as assessed by flow cytometry after Annexin V/DAPI double staining (Fig. 2c). IP2 derivative is thus more suitable for in vivo studies than the parent isoginkgetin.

To further assess the properties of the derivative IP2 in comparison with the parent isoginkgetin, we performed a RNA-seq followed by a differential gene expression analysis on MCA205 and B16F10 cells treated for 18 h with 35 μM of IP2 or 15 μM of isoginkgetin. We observed that IP2 impacts the expression of a highly reduced number of genes compared to the parent natural product isoginkgetin, which may explain its absence of toxicity (Fig. 2d). In addition, we assessed the effect of IP2 on alternative splicing. We first showed by quantitative RT-PCR that IP2 treatment in MCA205 and B16F10 cells increases the amount of unspliced pre-mRNA containing the

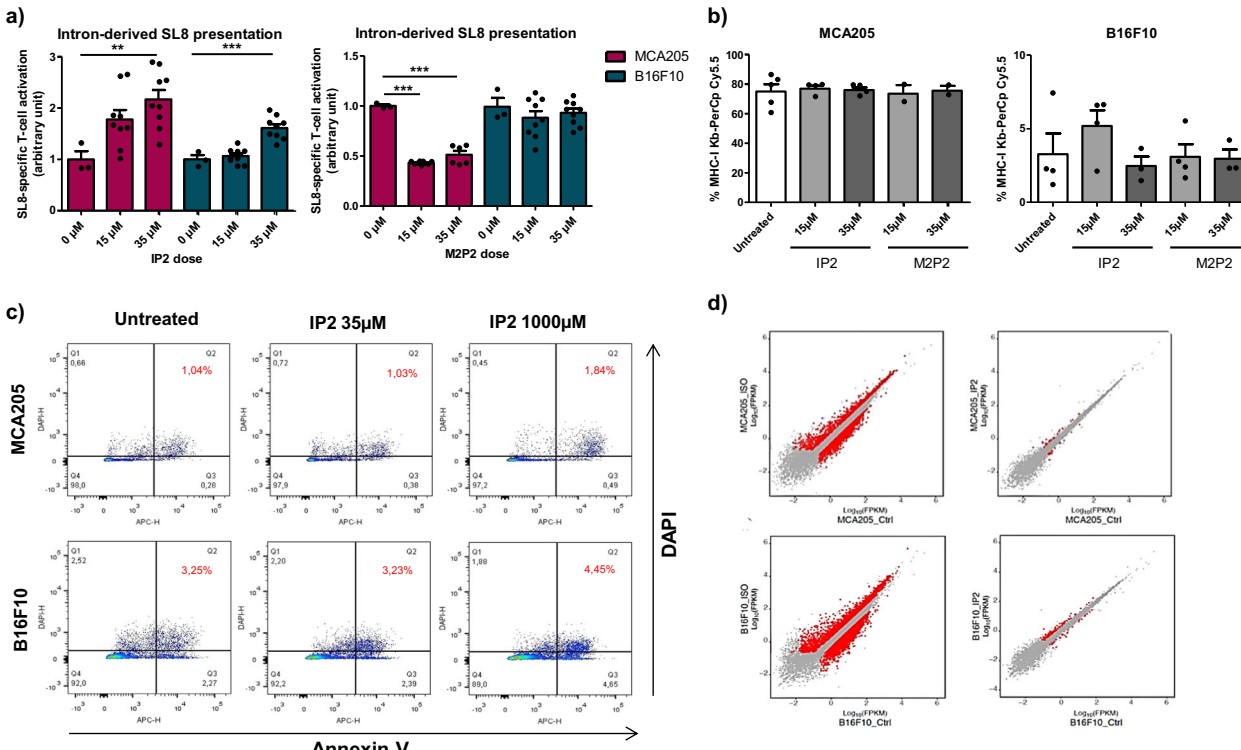

**Fig. 2 The derivative IP2 retains the properties of the parent isoginkgetin on antigen presentation with reduced cytotoxicity. a** SL8-specific B3Z T-cell activation after co-culture with MCA205 fibrosarcoma and B16F10 melanoma both transiently expressing the intron-derived SL8 epitope and treated for 18 h with IP2 (left panel) or M2P2 (right panel) at indicated doses. Unspecific B3Z T-cell activation was considered to normalize the results. Data are means ± SEM of relative B3Z T-cell activation for at least three biological replicates. **b** Analysis by flow cytometry of H-2K$^b$ expression at the surface of MCA205 fibrosarcoma (left panel) and B16F10 melanoma (right panel) treated with IP2 or M2P2 for 18 h at indicated doses. Data are means ± SEM of H-2K$^b$ frequency for three biological replicates. **c** Analysis by flow cytometry of the apoptotic status of MCA205 and B16F10 cells treated with 35 μM or 1000 μM of IP2 for 18 h and stained with DAPI and Annexin V-APC. Data highlight the proportion of late apoptotic cells (top right quadrant) and are representative of three independent experiments. **d** Comparison of gene expression profile between isoginkgetin-treated (left panels) and IP2-treated (right panels) MCA205 fibrosarcoma (top panels) and B16F10 melanoma (bottom panels). FPKM Fragments Per Kilobase per Million mapped fragments. **P < 0.01, ***P < 0.001 (One-way ANOVA).

globin-SL8-intron sequence, as isoginkgetin (Supplementary Fig. S3d). More generally, a differential splicing analysis performed from RNA-seq data revealed that isoginkgetin and IP2 induce similar but not identical splicing patterns in MCA205 and B16F10 (Supplementary Fig. S3e). In particular, about 30% of alternative splicing events (alternative 3' and 5' splice site, alternative first or last exon, mutually exclusive exons, intron retention or exon skipping) induced by IP2 in MCA205 cells are similarly induced by isoginkgetin on the same cells. This proportion reaches 70% in B16F10 cells.

All together, these results show that the compound IP2 is a water-soluble and non-toxic derivative of the natural product isoginkgetin. Importantly, the chemical modifications made to the original compound did not alter its positive effect on antigen presentation. Besides, IP2 and isoginkgetin are both splicing inhibitors but they may have different specificities of action.

**IP2 treatment reduces tumor growth and extends survival.** The derivative IP2, but not the derivative M2P2, enhances antigen presentation in vitro. We next assessed their antitumor activity in vivo. To this end, MCA205 and B16F10 tumor cells stably expressing the globin-SL8-intron construct or WT were subcutaneously inoculated into C57BL/6 mice as previously described. At day 5, 10, and 15 post-tumor inoculation, each group of

mice was intraperitoneally treated with 18 mg/kg of isoginkgetin, IP2 or M2P2. At this dose, a significant decrease in MCA205-globin-SL8-intron tumor growth was observed after treatment with IP2, slightly higher than that induced by isoginkgetin treatment, while no impact of M2P2 treatment was noticed (Fig. 3a and Supplementary Fig. S4a, left panels). In addition, the reduction in B16F10-globin-SL8-intron tumor growth was similar after treatment with 18 mg/kg of isoginkgetin or 18 mg/kg of IP2, while M2P2 had no effect (Fig. 3a and Supplementary Fig. S4a, right panels). Since IP2 is highly soluble in water, we tried to increase the dose injected into mice in order to achieve a greater reduction of tumor growth. Increasing the dose of IP2 to 36 mg/kg did improve the antitumor effect on B16F10-globin-SL8-intron but it has no effect on MCA205-globin-SL8-intron (Fig. 3a and Supplementary Fig. S4a). Strikingly, IP2 treatment slowed down the growth of both MCA205 WT and B16F10 WT tumors while neither isoginkgetin nor M2P2 treatments did (Fig. 3b and Supplementary Fig. S4b). More specifically, we observed a 40–60% decrease in MCA205 WT fibrosarcoma growth (Fig. 3b and Supplementary Fig. S4b, left panels) and a 40% decrease in B16F10 WT melanoma growth (Fig. 3b and Supplementary Fig. S4b, right panels) upon treatment with IP2. Furthermore, IP2 treatment extends overall survival with about 60% of mice bearing MCA205-globin-SL8-intron tumors that experience

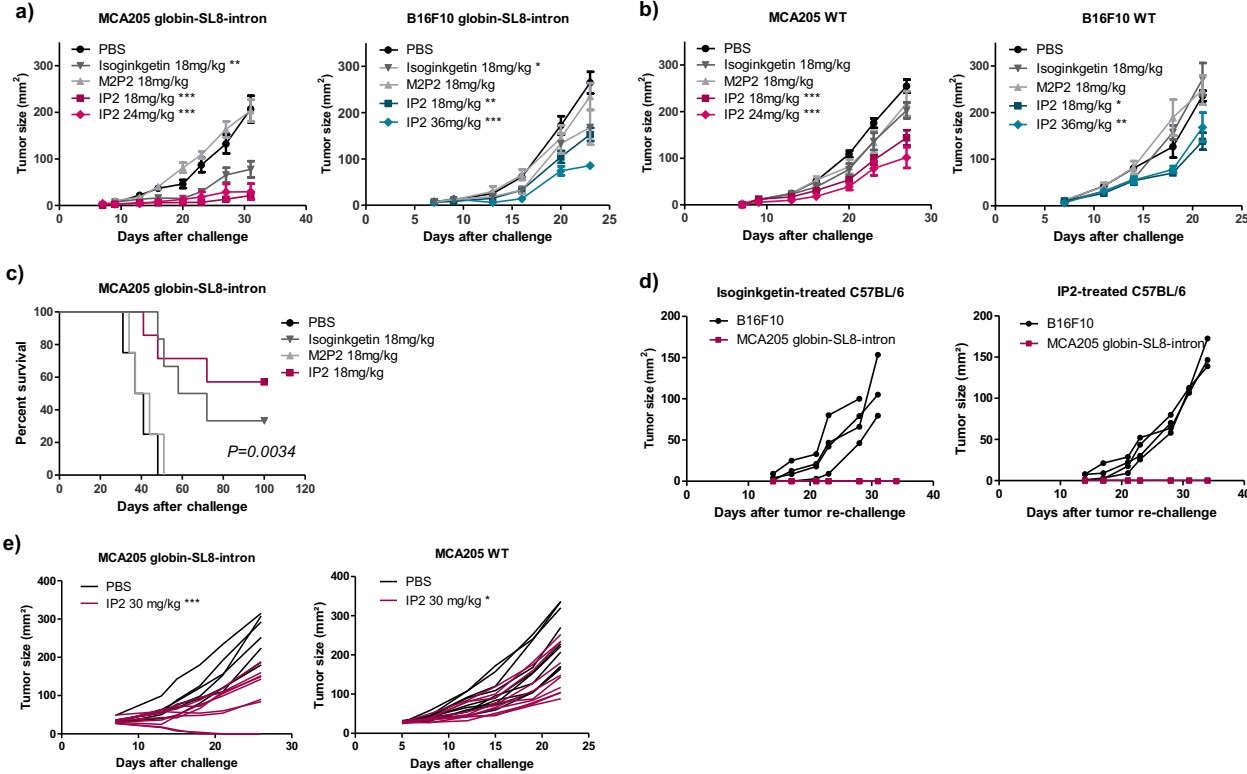

**Fig. 3 IP2 treatment reduces tumor growth and extends survival.** Tumor growth kinetics of (**a**) MCA205 fibrosarcoma (left panel) and B16F10 melanoma (right panel) both stably expressing the globin-SL8-intron construct and (**b**) MCA205 fibrosarcoma WT (left panel) and B16F10 melanoma WT (right panel) subcutaneously inoculated into the right flank of immunocompetent C57BL/6 mice thereafter treated at days 5, 10, and 15 with isoginkgetin, M2P2, or IP2 at indicated doses. Data are means ± SEM of tumor sizes for 6–8 mice per group. **c** Kaplan–Meier survival curves of mice subcutaneously challenged with MCA205 fibrosarcoma cells stably expressing the globin-SL8-intron construct and thereafter treated intraperitoneally with 18 mg/kg of isoginkgetin, 18 mg/kg of M2P2 or 18 mg/kg of IP2. A Log-rank (Mantel–Cox) test was performed. **d** Individual tumor growth kinetics of MCA205 cells stably expressing the globin-SL8-intron construct and B16F10 WT cells inoculated at day 100 in the ipsilateral side and the contralateral side, respectively, of C57BL/6 mice which experienced total tumor regression after first inoculation of MCA205-globin-SL8-intron at day 0 and subsequent treatment with isoginkgetin (left panel) or IP2 (right panel) at days 5, 10, and 15. **e** Individual tumor growth kinetics of MCA205 fibrosarcoma stably expressing the globin-SL8-intron construct (left panel) and MCA205 fibrosarcoma WT (right panel) subcutaneously inoculated into the right flank of immunocompetent C57BL/6 mice thereafter treated once established (30 mm²) at days 5, 10, and 15 with 30 mg/kg of IP2. *P < 0.05, **P < 0.01, ***P < 0.001 (One-way ANOVA).

complete tumor regression upon treatment with 18 mg/kg of IP2 versus 30% with 18 mg/kg of isoginkgetin (Fig. 3c).

Mice that were cured after treatment with IP2 or isoginkgetin were rechallenged with tumor cells about 3 months after first injection. They received MCA205-globin-SL8-intron cells in the ipsilateral flank and B16F10 WT cells in the contralateral flank. Without further treatment with one of the splicing inhibitors, mice did not develop MCA205-globin-SL8-intron tumors while B16F10 WT tumors grew over time (Fig. 3d). This result demonstrates that mice developed a long-term antitumor response against SL8-expressing tumors after treatment with isoginkgetin or IP2.

To further support the idea that IP2 has a strong antitumor activity, we evaluated the ability of the compound to treat established tumors. Mice were injected with either $2 \times 10^5$ MCA205-globin-SL8-intron or $2 \times 10^5$ MCA205 WT tumor cells and treatment was initiated when tumor size had reached 30 mm². Mice were then treated intraperitoneally with 30 mg/kg of IP2 every three days for a total of four injections. In that setting, we observed a 60% reduction in MCA205-globin-SL8-intron tumor size at day 26 after challenge (Fig. 3e and Supplementary Fig. S4c, left panels) and a 35% reduction in MCA205 WT tumor size at day 22 after challenge (Fig. 3e and Supplementary Fig. S4c, right panels).

Overall, these results suggest a correlation between the increase in PTP-derived antigen presentation observed in vitro and the reduction of the tumor growth in vivo upon treatment. Interestingly, unlike isoginkgetin, IP2 treatment slows down the growth of tumors that do not bear the highly immunodominant SL8 epitope. This difference of efficacy between the two molecules can be due to their different biodisponibility, which should be higher for the water-soluble IP2 molecule than for the hydrophobic isoginkgetin. We suggest that the derivative IP2 induces immunogenic epitopes at the surface of cancer cells driving the antitumor response.

**IP2 therapeutic effect is dependent on the adaptive immune response.** We have observed that IP2 therapeutic effect is stronger against tumors expressing the immunodominant SL8 epitope. Therefore, we assessed the requirement of an effective immune system and especially T cells for IP2 to exert its antitumor activity. MCA205 and B16F10 tumor cells stably expressing the globin-SL8-intron construct or WT were subcutaneously inoculated into Nu/Nu athymic nude mice that lack T cells but not B and NK cells. At day 5, 10, and 15 post-tumor inoculation, each group of mice was treated intraperitoneally with the dose of IP2 which was found to be the most efficient in immunocompetent mice. We observed that IP2 impacts the growth of none of the tumor models in immunodeficient nude mice (Fig. 4a, b). In addition, we further assessed the specific role of CD8+ T cells in IP2 action by depleting this population in vivo with an anti-CD8 T-cell monoclonal antibody. Mice were subcutaneously inoculated with MCA205-globin-SL8-intron or WT cells and thereafter treated every 3 days with depleting anti-CD8 T-cell antibody or with the isotype control. IP2 treatment was administered as previously at day 5, 10, and 15 post-tumor inoculation. Interestingly, CD8+ T-cell depletion completely abrogated the antitumor effect of the IP2 treatment (Fig. 4c). Therefore, this result confirms that the effect of IP2 treatment on tumor growth is dependent on the CD8+ T-cell response, which supports an antigen-driven cytotoxic activity against the tumor cells.

We then determined the effect of IP2 treatment on the CD8+ T-cell population in vivo. For this purpose, SL8-specific OT-1 T cells were adoptively transferred into C57BL/6 mice that were previously injected with MCA205 tumor cells stably expressing the globin-SL8-intron construct and treated twice with 24 mg/kg of IP2. Three days after the adoptive transfer, proliferation and expansion of SL8-specific CD8+ T cells was measured by flow cytometry using a SL8/H-2K$^b$ dextramer. We observed a two-fold increase in SL8-specific CD8+ T cells in mice that had been treated with IP2 prior to adoptive transfer of OT-1 cells (Fig. 4d) suggesting that IP2 can induce in vivo the proliferation of CD8+ T cells directed against tumor antigens.

**IP2 treatment alters the MHC-I immunopeptidome of tumor cells.** To confirm that the enhanced immune response observed upon IP2 treatment results from a change in cancer cell immunogenicity, we analyzed the repertoire of peptides displayed at the surface of cancer cells in MHC-I molecules. MCA205 WT fibrosarcoma were cultured alone or with IP2 and the peptides eluted from H-2K$^b$ and H-2D$^b$ molecules after immunoprecipitation were sequenced by mass spectrometry. The peptides identified in MCA205 untreated or treated cells showed a conventional length distribution with 8mers and 9mers being predominant in H-2K$^b$ enriched samples and 9mers making most of the peptides in H-2D$^b$ enriched samples (Fig. 5a). Unsupervised alignment of peptides sequences using the GibbsCluster server[33] also revealed typical binding motifs for H-2K$^b$ and H-2D$^b$ epitopes (Supplementary Fig. S5a and S5b). However, IP2 treatment altered the MHC-I immunopeptidome of MCA205 fibrosarcoma. Indeed, we identified epitopes whose presence at the cell surface was affected by IP2 treatment, whether positively or negatively (Supplementary data 1-4). Besides, we found epitopes absent from the surface of untreated MCA205 fibrosarcoma but identified after treatment with IP2. The peptide affinity for MHC-I molecules was estimated using the NetMHC 4.0 Server[34,35] and plotted against the fold change between untreated and treated cells (Fig. 5b). Twelve predicted binders of H-2K$^b$ molecules (affinity below 500 nM) and two predicted binders of H-2D$^b$ molecules (affinity below 500 nM) were found to be enriched at the surface of MCA205 fibrosarcoma upon treatment with IP2. To refine our selection, we used the IEDB Analysis Resource[36] to compute the immunogenicity score of the predicted strong binders, which accounts for peptide preferences in T-cell recognition. Eventually, two H-2K$^b$ epitopes with an affinity for MHC-I molecules lower than 500 nM and a predicted immunogenicity score above 0.1 were shown to be enriched in at least two out of three IP2-treated samples, the SGYEFIHKL 9mer from the tyrosine–tRNA ligase (UniProtKB - E9PX65) and the TNQDFIQRL 9mer (TL9) from the nischarin (UniProtKB–B7ZN33), which has been described as a tumor suppressor in breast and ovarian cancer (Fig. 5c)[37].

**IP2 treatment and a prophylactic cancer vaccine targeting the nischarin-derived epitope synergize to inhibit tumor growth.** Peptidomics enabled us to identify allegedly immunogenic epitopes overexpressed at the surface of MCA205 fibrosarcoma upon treatment with IP2. The nischarin-derived TNQDFIQRL epitope was further studied for the design of cancer vaccines. A 39-amino acid-long synthetic long peptide (TL9 SLP) encompassing the 9-amino acid-long sequence in central position was produced and its immunogenicity was assessed ex vivo. An irrelevant 39-amino acid-long SLP was also randomly generated and synthesized; both sequences are shared in Supplementary Fig. S6a. C57BL/6 mice were vaccinated with the TL9 SLP or the irrelevant SLP and one week later the frequency of IFN$_\gamma$-secreting CD8+ T cells in the spleens was assessed by ELISpot. We identified reactive CD8+ T cells against the TL9 SLP but not against the irrelevant SLP (Supplementary Fig. S6b). Significant concentrations of IL-2 were also detected by ELISA upon stimulation of the freshly isolated

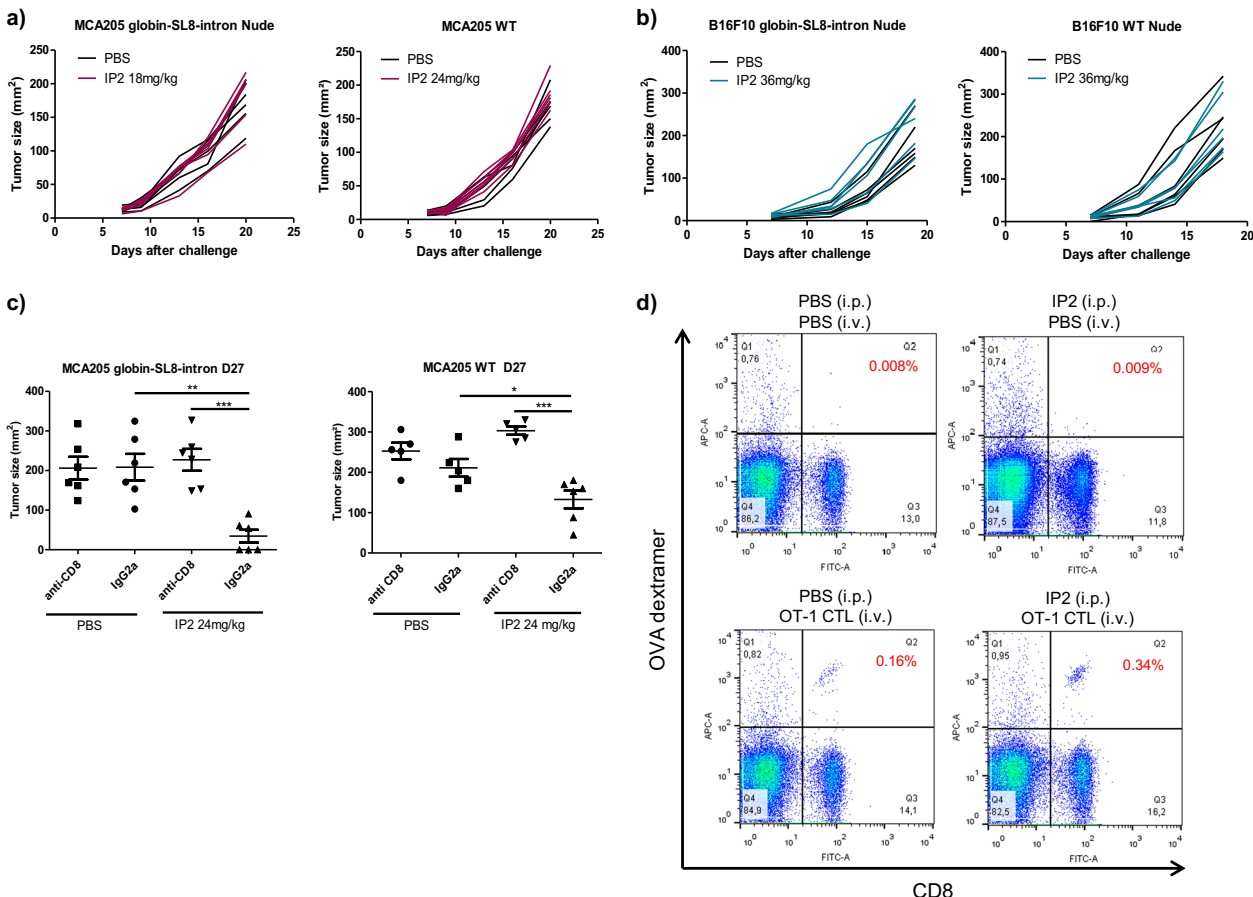

**Fig. 4 IP2 therapeutic effect is dependent on the adaptive immune response.** Individual tumor growth kinetics of (**a**) MCA205 fibrosarcoma stably expressing the globin-SL8-intron construct (left panel) and MCA205 fibrosarcoma WT (right panel) and (**b**) B16F10 melanoma stably expressing the globin-SL8-intron construct (left panel) and B16F10 melanoma WT (right panel) subcutaneously inoculated into the right flank of immunodeficient nude mice thereafter treated at days 5, 10, and 15 with 18 mg/kg of IP2. **c** Tumor size at killing of MCA205 fibrosarcoma stably expressing the globin-SL8-intron construct (left panel) and MCA205 fibrosarcoma WT (left panel) subcutaneously inoculated into the right flank of immunocompetent C57BL/6 mice thereafter treated with 24 mg/kg of IP2 at days 5, 10, and 15 and CD8+ T-cell depleting antibody every three days. Data are given as mean ± SEM of tumor size for six mice per group. **d** Analysis by flow cytometry of SL8-specific OT-1 CD8+ T-cell proliferation after intravenous adoptive transfer of OT-1 CD8+ T cells into immunocompetent C57BL/6 mice bearing MCA205-globin-SL8-intron fibrosarcoma and treated with 24 mg/kg of IP2. Data highlight the frequency of SL8-specific OT-1 CD8+ T cells (top right quadrant) in the spleens and are representative of five mice per group. *$P < 0.05$, **$P < 0.01$, ***$P < 0.001$ (One-way ANOVA).

CD8+ T cells with BMDCs pulsed with the TL9 SLP but not the irrelevant SLP. Here, the IFNγ ELISpot and the IL-2 ELISA assays showed closely correlated results (Supplementary Fig. S6c).

The TNQDFIQRL epitope shows a good computed immunogenicity score and the TL9 SLP was shown to be immunogenic in two different immunoassays therefore the antitumor potential of the combination between IP2 and the TL9 SLP was assessed in vivo. In a prophylactic setting, mice were vaccinated twice with 100 μg of TL9 SLP, then challenged subcutaneously with MCA205 fibrosarcoma and finally treated intraperitoneally with 24 mg/kg of IP2 (Fig. 6a). Vaccination with the TL9 SLP without consecutive treatment with IP2 had no effect on the growth of MCA205 tumors (Fig. 6b). However, tumor growth at day 24 after challenge was inhibited by 35% with IP2 alone while the combination of IP2 with the TL9 SLP induced a 60% tumor growth reduction (Fig. 6c) and led to a prolonged overall survival (Fig. 6d). The antitumor effect of the TL9 SLP combined with IP2 was also assessed in a therapeutic setting. Mice were subcutaneously challenged with MCA205 cells and thereafter injected concomitantly with the TL9 SLP and/or 24 mg/kg of IP2 (Fig. 7a). We observed results comparable to those obtained in the prophylactic setting. Although some mice injected with the TL9

SLP experienced limited tumor progression (Fig. 7b), the vaccination alone has no significant effect on tumor size at day 22 after challenge (Fig. 7c). In contrast, when combined with IP2, the TL9 SLP induced a 60% tumor growth reduction. Eventually, mice treated with the combination also benefited from a prolonged overall survival (Fig. 7d).

Overall, we show here in the mouse MCA205 fibrosarcoma model that IP2 induces the presentation of immunogenic epitopes that can be targeted in cancer vaccines. This suggests the use of IP2 to build new epitope-based cancer immunotherapies.

## Discussion
Splicing abnormalities have emerged as a specific feature of cancer cells and are studied as predictive markers for patient survival[38,39]. Tumors bearing spliceosome alterations are key targets for treatments with splicing inhibitors some of which are currently under development in acute myeloid leukemia[40]. Here we observed that treatment with the splicing inhibitor isoginkgetin or its derivative IP2 reduces the growth of MCA205 fibrosarcoma and B16F10 melanoma bearing the PTP-derived SL8 epitope and significantly extends mice survival (Fig. 3a–c).

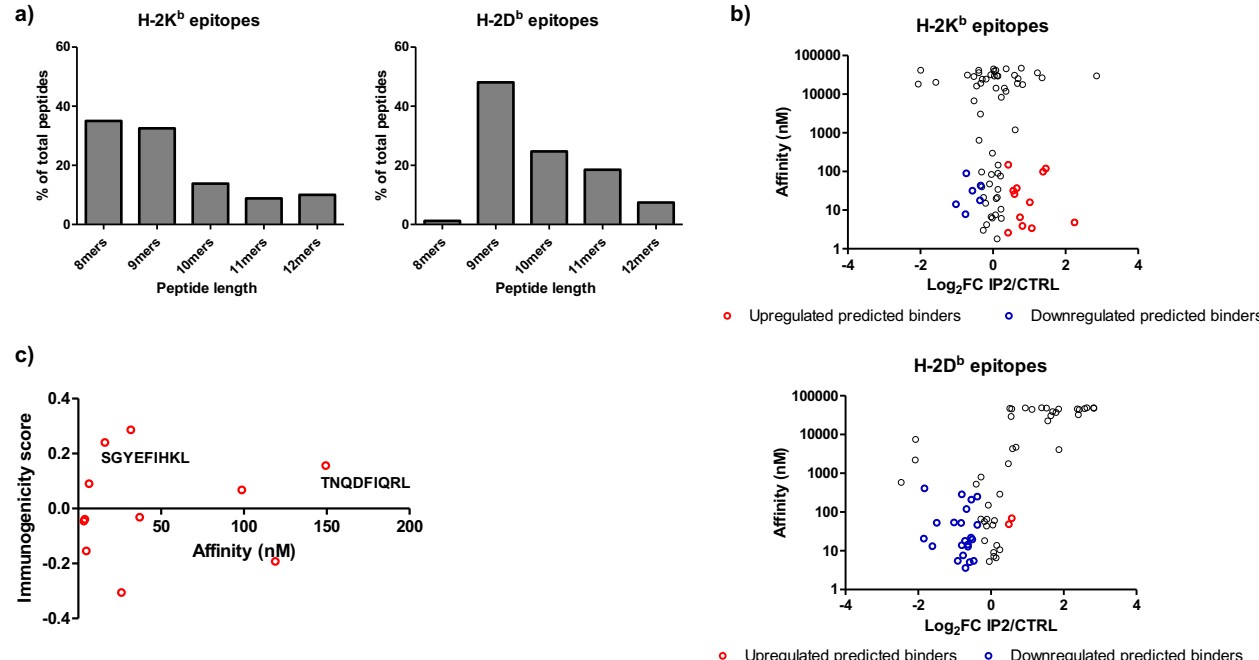

**Fig. 5 IP2 alters the MHC-I immunopeptidome of tumor cells. a** Length distribution of epitopes identified at the surface of MCA205 cells in H-2K$^b$ molecules (left panel) and H-2D$^b$ molecules (right panel) after immunoprecipitation. **b** Affinity for MHC-I molecules of peptides identified after H-2K$^b$ immunoprecipitation (upper panel) and H-2D$^b$ immunoprecipitation (lower panel). Red dots represent the predicted binders (affinity < 500 nM) displaying a fold change above 1.2 between IP2-treated vs untreated replicates. Blue dots represent the predicted binders (affinity < 500 nM) displaying a fold change lower than 0.8 between IP2-treated vs untreated replicates. **c** Computed immunogenicity score of predicted binders of H-2K$^b$ molecules upregulated after treatment with IP2. Epitopes were labelled if the immunogenicity score was above 0.1 and their presence validated in at least two out of three replicates.

We also noticed that mice which experienced complete tumor regression upon treatment with IP2 developed a long-lasting immune response (Fig. 3d). Importantly, the antitumor activity of IP2 was completed abolished in immunodeficient nude mice or after in vivo depletion of CD8$^+$ T cells in immunocompetent C57BL/6 mice (Fig. 4a, b). Moreover, IP2 treatment was shown to reshape the MHC-I immunopeptidome, leading to a new antigen repertoire at the cell surface. In fact, we identified peptides enriched at the surface of MCA205 fibrosarcoma upon IP2 treatment and peptides absent from untreated cells that appeared upon treatment (Supplementary data 1-4). In particular, the IP2-induced nischarin-derived epitope TNQDFIQRL (TL9) was shown to be immunogenic and to induce tumor growth defect in vivo when targeted in peptide vaccines in combination with IP2 (Figs. 6 and 7). We highlight here that splicing inhibitors can act as immunomodulatory agents which action is mediated by the adaptive immune system while they have been only exploited so far for their cytotoxicity directed toward tumor cells bearing alterations of their splicing machinery.

Splicing inhibitors isoginkgetin, spliceostatin A and pladienolide B have been shown to promote pre-mRNA accumulation in the nucleus[26,41,42]. However, no link has been made between the accumulation of pre-mRNAs and the observed increase in antigen presentation so far. We propose that the accumulation of pre-mRNA in the nucleus provides more templates for nuclear translation leading to an enrichment of PTP-derived antigens used for direct MHC-I antigen presentation. Besides, we showed that the increased presentation of the PTP-derived SL8 peptide observed upon treatment with both isoginkgetin and IP2 relies on the alteration of the splicing mechanism since the drugs has no effect on the cells expressing the SL8-encoding cDNA (Fig. 1c). Such regulation of the MHC-I antigen presentation pathway by splicing events have been described before, for example for the EBNA-3-derived FLR epitope[43]. Interestingly, the inhibition of the splicing machinery not only alters the presentation of intron-derived epitopes but also exon-derived ones. Indeed, IP2 enhanced the presentation of the SL8 epitope in cancer cells expressing the globin-SL8-exon construct (Fig. 1b) and the endogenous nischarin-derived TL9 epitope in MCA205 fibrosarcoma. That could be the result of the accumulation of exonic sequences in the pre-mRNA or, in other cases, the induction of a frameshift leading to the translation of an exon out-of-frame.

IP2 is not cytotoxic for cancer cells (Supplementary Fig. S3) and does not induce apoptosis at the highest tested dose (Fig. 2c) thus it is unlikely that IP2 would be an inducer of immunogenic cell death[44]. Actually, we showed with the SL8/OT-1 T-cell model that the overexpression of the SL8 epitope upon treatment with IP2 promoted the proliferation of SL8-specific T cells and thus the elimination of SL8-expressing tumors (Fig. 4). Conversely, the isoginkgetin derivative M2P2, which does not increase the presentation of the PTP-derived SL8 epitope does not prevent the growth of SL8-expressing tumors in vivo (Figs. 2a, 3a and 3b). Moreover, we noticed a greater effect of IP2 on SL8-expressing tumors than WT tumors with a complete tumor regression in a significant number of mice. The chicken ovalbumin-derived SL8 epitope is highly immunogenic and not subject to thymic selection in mice, therefore we can reasonably hypothesize that its presentation by the tumor drives the adaptive immune response. In particular, the lysis of SL8-expressing tumor cells by CTLs can turn the surrounding stroma cells into antigen presenting cells thus promoting the "bystander" elimination of antigen loss variants (ALVs) that no longer express the SL8 epitope[32]. The SL8-driven immune response could also spread to other antigens and prevent tumors from escaping eradication by CTLs[45]. It is then not surprising that we observed a resumption of tumor growth in vivo once mice were not subjected to IP2 treatment anymore, in most experiments 15–17 days after challenge.

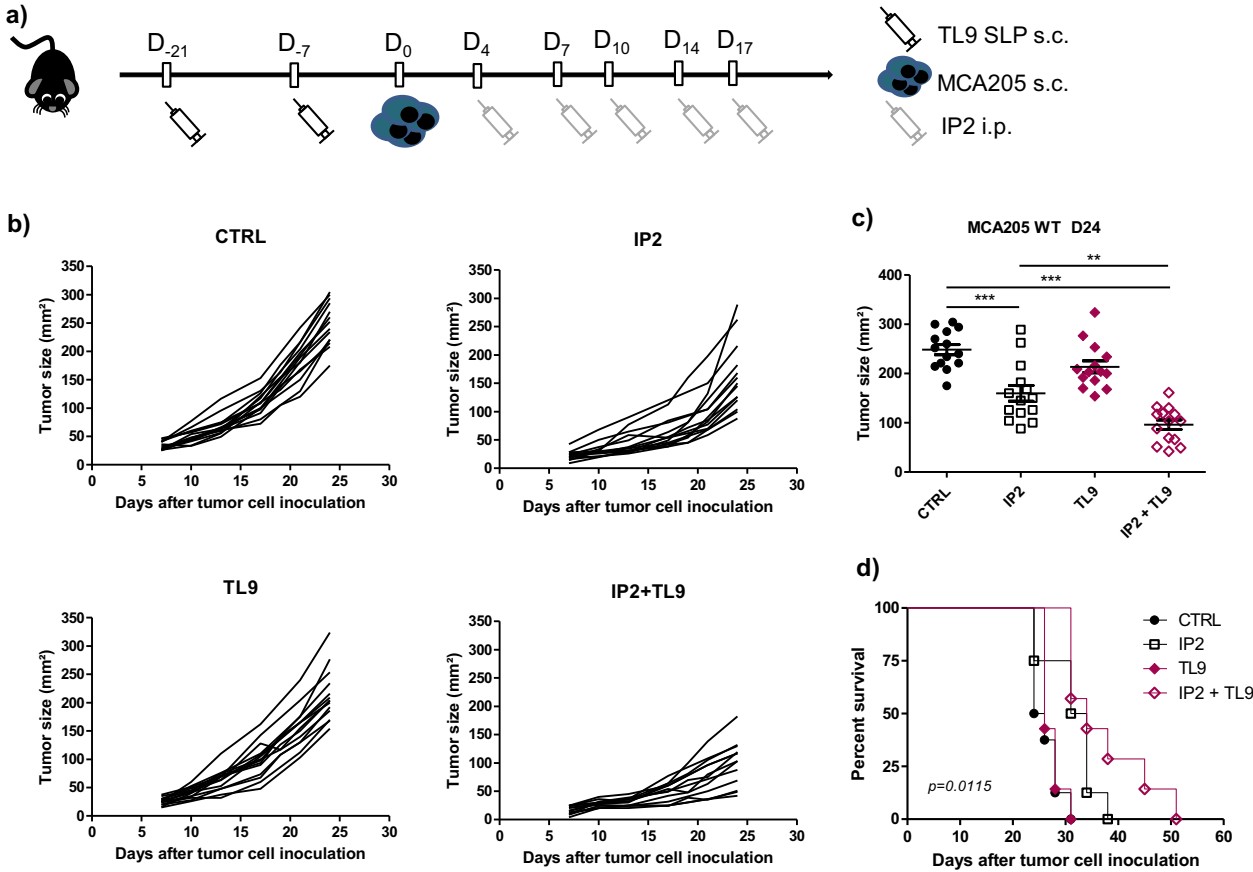

**Fig. 6 IP2 treatment and a prophylactic cancer vaccine targeting the nischarin-derived epitope synergize to inhibit tumor growth. a** Experimental setting for the evaluation of the antitumor activity of IP2 in combination with the prophylactic TL9 SLP vaccine. **b** Individual tumor growth kinetics and (**c**) tumor size at killing of MCA205 fibrosarcoma subcutaneously inoculated into the right flank of immunocompetent C57BL/6 mice previously vaccinated with the TL9 SLP and thereafter treated with 24 mg/kg of IP2 from tumor cell inoculation. **d** Kaplan–Meier survival curves of MCA205 fibrosarcoma-bearing immunocompetent C57BL/6 mice vaccinated in a prophylactic setting with the TL9 SLP and thereafter treated with 24 mg/kg of IP2 from tumor cell inoculation. Log-rank (Mantel–Cox) test was performed for survival analysis. **$P < 0.01$, ***$P < 0.001$ (One-way ANOVA).

Importantly, IP2 treatment slows down the growth of WT fibrosarcoma and melanoma, which suggests that an immuno-dominant epitope is not required to observe an impact on tumor growth but however necessary to observe complete tumor regression (Fig. 4). While this could be explained by a quantitative change of endogenous epitopes in WT tumors upon treatment, a qualitative change may also occur. The plasticity of the immunopeptidome has been demonstrated in several studies[46], and has been linked to many parameters such as intracellular metabolism, transcriptional regulations, gene expression and genetic polymorphism[47–50]. Several mass spectrometry-based peptidomic analysis have also highlighted the modulation of the MHC-I immunopeptidome upon various treatments. For example, it has been reported that the antimetabolite gemcitabine[51] and decitabine, the TNFα and $IFN_\gamma$ pro-inflammatory cytokines[52] or the rapamycin (inhibitor of the mTOR kinase)[1] have the capacity to modify quantitatively and qualitatively the repertoire of epitopes presented at the surface of treated cells. We show here that interfering with the spliceosome using the IP2 compound leads to a remodeling of the MHC-I immunopeptidome of MCA205 fibrosarcoma (Supplementary data 1-4). This IP2-induced immunopeptidome is composed not only of epitopes whose expression was modulated upon treatment but also of epitopes that appear upon treatment with IP2. The plasticity of the MHC-I immunopeptidome of non-conventional antigens upon IP2 treatment has not been defined yet and is worth further investigation. We can expect splicing inhibitors to boost the

presentation of non-conventional epitopes generated from introns, 5′ and 3′ untranslated regions or exon–intron junctions[53,54]. In particular, the isoginkgetin and IP2 compounds were shown to induce intron retention in MCA205 and B16F10 cells in our differential splicing analysis and intron retention is a source of neoepitopes in cancer[55]. We contemplate the use of IP2 to enlarge the pool of targetable tumor-associated (TAAs) and tumor-specific (TSAs) antigens from different regions of the genome in the context of epitope-based immunotherapies.

The exon-derived TL9 epitope from the tumor suppressor nischarin was shown to be immunogenic and to synergize with IP2 in cancer vaccines to inhibit the growth of MCA205 fibrosarcoma. Recent studies highlighted the role of nischarin in the regulation of breast tumorigenesis[56] related to the secretion of exosomes from breast cancer cells[57]. The nischarin-derived TL9 epitope has been identified throughout multiple proteomic studies referenced in the Immune Epitope Database (IEDB) but has never been further exploited so far. Importantly, the TL9 epitope was found to be expressed in healthy tissues[58] however we observed neither an antitumor effect nor any deleterious auto-immune response in mice injected with the TL9 SLP alone. This suggests that the basal level of expression of the antigen at the surface of MCA205 fibrosarcoma is not sufficient to allow CTL recognition, which is no more the case after IP2 treatment. The 39-amino-acid-long TL9 SLP was generated by translating the corresponding sequence identified in the RNA-seq analysis of MCA205 fibrosarcoma. Interestingly, we identified a point

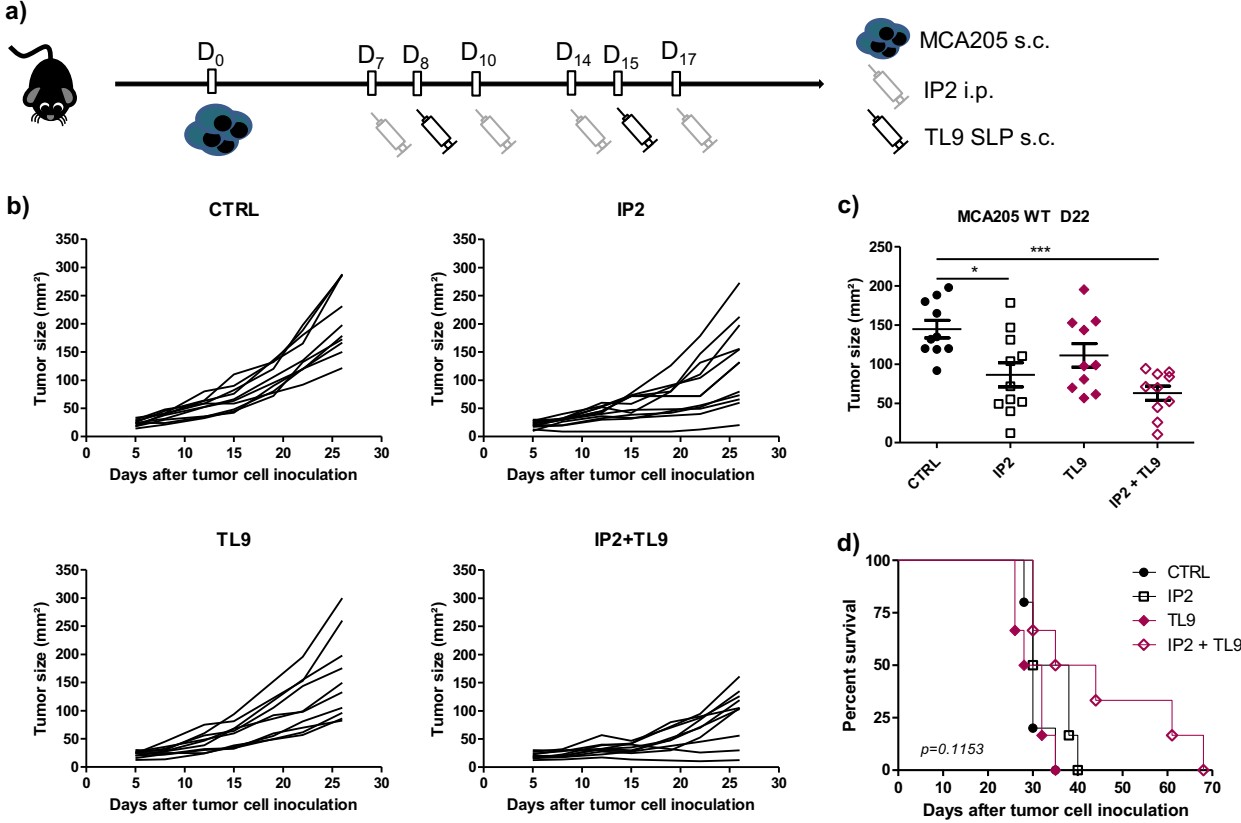

**Fig. 7 IP2 treatment and a therapeutic cancer vaccine targeting the nischarin-derived epitope synergize to inhibit tumor growth. a** Experimental setting for the evaluation of the antitumor activity of IP2 in combination with the therapeutic TL9 SLP vaccine. **b** Individual tumor growth kinetics and (**c**) tumor size at killing of MCA205 fibrosarcoma subcutaneously inoculated into the right flank of immunocompetent C57BL/6 mice and thereafter injected once a week with the TL9 SLP and twice a week with 24 mg/kg of IP2. **d** Kaplan–Meier survival curves of MCA205 fibrosarcoma-bearing immunocompetent C57BL/6 treated concomitantly with the TL9 SLP and 24 mg/kg of IP2 for 2 weeks. Log-rank (Mantel–Cox) test were performed for survival analysis. *$P < 0.05$, ***$P < 0.001$ (One-way ANOVA).

mutation in the cancer cell transcriptome, which translates into an amino acid switch, from Alanine (A) to Aspartic acid (D), at the position preceding the epitope. The flanking regions of the epitopes in synthetic long peptides can be optimized to promote epitope generation and presentation[59], we hypothesize that this mutation found in the nischarin gene in MCA205 cells could potentially play a role in the increase presentation of the epitope compared with other healthy tissues. Future work will be needed to explore the effect of IP2 treatment on the presentation of the TL9 epitope with and without this mutation within the flanking regions, as well as other IP2-induced MHC ligands in cancerous and noncancerous cells.

Eventually, this work suggests a role for splicing regulation in the generation of MHC-I peptide for cancer immunosurveillance. The frequent mutations affecting the spliceosome in cancer cells may promote their evasion from T cells and NK cells. This raises the potential of the spliceosome as a druggable target in cancer immunotherapy. Potential side effects inherent to the alteration of such essential cellular mechanisms must be considered in the light of the benefice/risk balance. The use of proteasome inhibitors for myeloma or lymphoma treatment is one pertinent example[60]. We reaffirm here that we did not observe any toxicity related to the use of the IP2 compound in vivo. While a deeper understanding of isoginkgetin and IP2 effects on splicing is needed, their positive impact on the immune response opens a large field of study on the link between splicing regulation and MHC-I immunopeptidome regulation. This could lead to the development of innovative anticancer immunotherapies targeting the spliceosome.

## Methods

**Cell culture, plasmid DNA transfection, drugs, and peptides**. MCA205 mouse fibrosarcoma cell line was cultured at 37 °C under 5% $CO_2$ in RPMI 1640 medium (Life Technologies) in the presence of 1% glutamine, 1% sodium pyruvate, 1% non-essential amino-acids, 1% penicillin/streptomycin and 10% FBS (Life Technologies). B16F10 mouse melanoma cell line was cultured at 37 °C under 5% $CO_2$ in DMEM medium (Life Technologies) containing 1% glutamine, 1% penicillin/streptomycin and 10% FBS. The SL8/H-2$K^b$-specific B3Z CD8$^+$ T-cell hybridoma[61] were cultured at 37 °C under 5% $CO_2$ in RPMI 1640 medium supplemented with 1% glutamine, 0.1% 2-mercaptoethanol, 1% penicillin/streptomycin, and 10% FBS. Once a month, mycoplasma contamination in cell cultures was assessed using the Venor®GeM OneStep mycoplasma detection kit (Minerva biolabs). All cells were used within four weeks after thawing (≈10 passages). MCA205 cells were transfected with the reagent jetPRIME (Ozyme) according to the manufacturer protocol. B16F10 cells were transfected with the reagent GeneJuice (Millipore) according to the manufacturer protocol. The plasmids YFP-Globin-SL8-intron, YFP-Globin-SL8-exon and PCDNA3 have been described previously[17]. The pre-mRNA splicing inhibitor isoginkgetin (Merck Millipore) was resuspended in DMSO. The IP2 and M2P2 derivatives were synthesized from the isoginkgetin powder and resuspended in PBS. Synthetic long peptides (39AA) were synthesized by GL Biochem, resuspended at 5 mg/mL in PBS at delivery and stored at −20 °C before use.

**Procedures and characterizations of compounds 1, 2, IP2, and M2P2**. Solvents and reagents are obtained from commercial suppliers and were used without further purification. Analytical TLC was performed using Merck silica gel F254 (230–400 mesh) plates and analyzed by UV light. For silica gel chromatography, the flash chromatography technique was used, with Merck silica gel 60 (230–400 mesh) and p.a. grade solvents unless otherwise noted. The $^1$H NMR, $^{31}$P NMR, and $^{13}$C NMR spectra were recorded in either CDCl$_3$, D$_2$O, or DMSO-d6 on Bruker Avance 300 spectrometers, or 200 spectrometers. The chemical shifts of $^1$H and $^{13}$C are reported in ppm relative to the solvent residual peaks. IR spectra were measured on a Bruker Vector 22 spectrophotometer. Melting points were recorded on a Büchi B-450 apparatus and are uncorrected. High resolution mass spectra

(HRMS) were recorded on a MicroMass LCT Premier TOF ESI Spectrometer. For the synthesis of compound 1 (Supplementary Fig. S2), KOH (445 mg, 7.92 mmol, 8 eq.) was added to a suspension under argon of isoginkgetin (561 mg, 0.99 mmol, 1 eq.) in water (560 mL). The mixture was stirred at room temperature for 15 min, and was added successively dichloromethane (560 mL), tetrabutylammonium bromide (638 mg, 1.98 mmol, 2 eq.), and diethylchlorophosphate (376 mg, 316 µl, 2.18 mmol, 2.2 eq.). The reaction medium was stirred at room temperature for 2 h and after separation of the phases, the organic layer was dried over sodium sulfate, filtered and concentrated under reduced pressure. The crude product was purified on a silica gel column ($CH_2Cl_2$/MeOH/$HCl_{aq}$(1 N) :100:0:0 to 99:1:0.1) to yield 635 mg of compound 1 as a yellow powder (644 mg, 0.76 mmol, 77% yield). $^1$H NMR (300 MHz, $CDCl_3$) $\delta$: 13.00 (s, 1H, OH), 12.79 (s, 1H, OH), 8.02 (dd, J = 8.7, 2.4 Hz, 1H, ArH), 7.97 (d, J = 2.4 Hz, 1H, ArH), 7.46 (d, J = 8.9 Hz, 2H, ArH), 7.18 (d, J = 8.8 Hz, 1H, ArH), 6.99 (1H, ArH), 6.95 (1H, ArH), 6.84 (d, J = 8.9 Hz, 2H, ArH), 6.65 (dd, J = 12.3, 5.1 Hz, 3H, ArH), 4.32–4.18 (m, 4H, $CH_2$), 4.11–3.95 (m, 4H, $CH_2$), 3.81 (m, 6H, $OCH3$), 1.37 (t, J = 7.0 Hz, 6H, $CH_3$), 1.22 (td, J = 6.7, 4.0 Hz, 6H, $CH_3$). $^{13}$C NMR (75 MHz, $CDCl_3$) $\delta$: 183.0 (CO), 164.4 (Ar), 162.9 (Ar), 162.3 (Ar), 161.9 (Ar), 161.1 (Ar), 157.1 (Ar), 156.1 (Ar), 156.0 (Ar), 154.4 (Ar), 153.6 (Ar), 153.5 (Ar), 131.2 (ArH), 128.6 (ArH), 127.9 (2ArH), 123.1 (Ar), 123.1 (Ar), 121.5 (Ar), 114.7 (2ArH), 111.3 (ArH), 108.3 (ArH), 104.9 (ArH), 104.3 (ArH), 103.9 (ArH), 103.9 (ArH), 103.7 (ArH), 99.1 (ArH), 99.0 (ArH), 65.2 (d, J = 6.0 Hz, $2CH_2$), 64.9 (d, J = 6.3 Hz, $2CH_2$), 56.1 ($OCH_3$), 55.6 ($OCH_3$), 16.3 ($CH_3$), 16.2 ($CH_3$), 16.1 ($CH_3$). 31 P NMR (81 MHz, $CDCl_3$) $\delta$: −7.30, −7.50. IR (neat): 2989 (CH), 1653 and 1604 (CO), 1582 $cm^{-1}$. HRMS (ESI-TOF) m/z; [M + H] + Calcd. for $C_{40}H_{41}O_{16}P_2$ 839.1870; found 839.1865. Melting point = 155 °C. For the synthesis of IP2 (see supplementary Fig. S2), TMSI (2.2 mL; 16.38 mmol; 9 eq.) was added to a solution under argon of compound 1 (1.53 g; 1.82 mmol; 1 eq.) in anhydrous dichloromethane (36 mL) and the reaction mixture was stirred at room temperature for 16 h. After evaporation of the solvent under reduced pressure, the yellow solid (2.07 g) was dissolved in anhydrous methanol (60 mL) and NaOH (291 mg; 7.28 mmol; 4 eq.) was added. The solution was stirred at room temperature for 30 min and the solvent was evaporated under reduced pressure to yield IP2 (1.43 g, 1.74 mmol; 96%) as a yellow solid. $^1$H NMR (300 MHz, $D_2O$) $\delta$: 7.64 (brs, 1H, ArH), 7.51 (brs, 1H, ArH), 7.24 (s, 1H, ArH), 7.00 (brs, 3H, ArH), 6.81 (d, J = 7.0 Hz, 2H, ArH), 6.58–6.41 (m, 3H, ArH), 6.33 (brs, 1H, ArH), 3.83 (s, 3H, $CH_3$), 3.63 (s, 3H, $CH_3$). 13 C NMR (75 MHz, $D_2O$) $\delta$: 182.5 (CO), 164.5 (Ar), 161.8 (Ar), 160.9 (Ar), 160.8 (Ar), 160.4 (Ar), 159.4 (Ar), 159.3 (Ar), 158.6 (Ar), 156.9 (Ar), 153.7 (Ar), 130.9 (ArH), 128.3 (ArH), 127.3 (2ArH), 122.5 (Ar), 122.1 (Ar), 121.5 (Ar), 114.0 (2ArH), 111.8 (ArH), 107.2 (Ar), 107.1 (Ar), 105.5 (Ar), 104.9 (Ar), 103.6 (ArH), 103.3 (Ar), 102.6 (ArH), 99.8 (ArH), 99.7 (ArH), 56.1 ($OCH_3$), 55.4 ($OCH_3$). $^{31}$P NMR (81 MHz, $D_2O$) $\delta$: −0.49, −1.59. IR (neat): 3423 (broad), 2989 (CH), 1650 and 1600 (CO), 1563 $cm^{-1}$. HRMS (ESI-TOF) m/z; [M-H]- Calcd. for $C_{32}H_{19}O_{16}P_2$ 725.0467; found 725.0466. Melting point >350 °C. HPLC purity could not be determined due to the hydrolysis of phosphate during analysis. For the synthesis of compound 2, $K_2CO_3$ (90 mg, 0.65 mmol, 5 eq.) and iodomethane (24 µl, 0.39 mmol, 3 eq.) were successively added to a solution under argon of compound 1 (109 mg; 0.13 mmol; 1 eq.) in anhydrous DMF (1.5 mL). The mixture was stirred at room temperature for 20 h and filtered. The solid was washed with ethyl acetate and the filtrate was washed with brine (3 × 30 mL). The organic layer was dried over sodium sulfate, filtered and concentrated under reduced pressure to yield 110 mg of compound 2 as a yellow powder (109 mg, 0.126 mmol, 97% yield).$^1$H NMR (300 MHz, $CDCl_3$) $\delta$: 7.99 (dd, J = 8.8, 2.1 Hz, 1H, ArH), 7.93 (d, J = 2.2 Hz, 1H, ArH), 7.38 (d, J = 8.9 Hz, 2H, ArH), 7.16 (d, J = 8.7 Hz, 1H, ArH), 7.11 (s, 1H, ArH), 7.04 (s, 1H, ArH), 6.80 (d, J = 8.9 Hz, 2H, ArH), 6.67 (m, 3H, ArH), 4.31 – 4.18 (m, 4H, $CH_2$), 4.16–3.90 (m, 10H, $CH_2$ and $OCH_3$), 3.78 (d, J = 3.3 Hz, 6H, $OCH_3$), 1.36 (t, J = 7.1 Hz, 6H, $CH_3$), 1.22 (m, 6H, $CH_3$). 13 C NMR (75 MHz, $CDCl_3$) $\delta$: 177.9 (CO), 177.4 (CO), 162.3 (Ar), 161.2 (Ar), 161.0 (Ar), 160.9 (Ar), 160.5 (Ar), 160.4 (Ar), 158.9 (Ar), 156.4 (Ar), 154.8 (Ar), 154.8 (Ar), 152.3 (Ar), 152.2 (Ar), 130.7 (ArH), 128.1 (ArH), 127.5 (2ArH), 123.4 (Ar), 123.3 (Ar), 121.6 (Ar), 114.5 (2ArH), 112.0 (Ar), 111.1 (ArH), 108.0 (ArH), 107.3 (ArH), 101.1 (ArH), 101.1 (ArH), 99.9 (ArH), 99.8 (ArH), 99.5 (ArH), 65.2 (d, J = 6.3 Hz, $2CH2$), 65.1 (d, J = 6.4 Hz, $CH2$), 64.9 (d, J = 6.6 Hz, $CH_2$), 56.9 ($OCH_3$), 56.8 ($OCH_3$), 56.0 ($OCH_3$), 55.5 ($OCH_3$), 16.2 ($CH_3$), 16.1 ($CH_3$), 16.0 ($CH_3$). $^{31}$P NMR (81 MHz, $CDCl_3$) $\delta$: −7.30, −7.50. IR (neat): 3492, 2984–2934 (CH), 1647 and 1605 (CO), 1573 $cm^{-1}$. HRMS (ESI-TOF) m/z; [M + H] + Calcd. for $C_{42}H_{45}O_{16}P_2$ 867.2177; found 867.2177. Melting point = 142 °C. For the synthesis of M2P2, TMSI (0.16 mL; 1.17 mmol; 9 eq.) was added to a solution under argon of compound 2 (109 mg; 0.13 mmol; 1 eq.) in anhydrous dichloromethane (8 mL) and the reaction mixture was stirred at room temperature for 16 h. After evaporation of the solvent under reduced pressure, the yellow solid was dissolved in anhydrous methanol (4 mL) and NaOH (21 mg; 0.52 mmol; 4 eq.) was added. The solution was stirred at room temperature for 30 min and the solvent was evaporated under reduced pressure to yield M2P2 (101 mg, 0.12 mmol, 95%) as a yellow solid. $^1$H NMR (300 MHz, $D_2O$) $\delta$: 7.81 (s, 1H, ArH), 7.71 (m, 1H, ArH), 7.41 (s, 1H, ArH), 7.23–6.93 (m, 4H, ArH), 6.81 (s, 1H, ArH), 6.54 (m, 3H, ArH), 6.33 (s, 1H, ArH), 4.05 (s, 3H, $OCH_3$), 3.94 (s, 3H, $OCH_3$), 3.70 (s, 3H, $OCH_3$), 3.54 (s, 3H, $OCH_3$). $^{13}$C NMR (75 MHz, $D_2O$) $\delta$: 179.9 (CO), 162.4 (Ar), 161.5 (Ar), 160.8 (Ar), 160.4 (Ar), 160.3 (Ar), 160.2 (Ar), 160.1 (Ar), 159.9 (Ar), 159.7 (Ar), 159.4 (Ar), 158.9 (Ar), 157.9 (Ar), 155.6 (Ar), 130.9 (ArH), 128.1 (ArH), 127.1 (2ArH), 126.7 (ArH), 122.8 (Ar), 122.1 (Ar), 113.8 (2ArH), 111.6 (ArH), 108.2 (Ar), 107.6 (ArH), 105.9 (ArH), 104.9 (ArH), 100.9

(ArH), 99.6 (ArH), 56.3 ($OCH_3$), 56.2 ($OCH_3$), 56.2 ($OCH_3$), 56.0 ($OCH_3$). $^{31}$P NMR (81 MHz, $CDCl_3$) $\delta$: −0.16, −0.80. IR (neat): 3393 (broad), 2984 (CH), 1631 and 1593 (CO), 1565 $cm^{-1}$. HRMS could not be determined due to the hydrolysis of phosphate during analysis Melting point >350 °C. HPLC purity could not be determined due to the hydrolysis of phosphate during analysis.

**T-cell activation assay.** MCA205 fibrosarcoma and B16F10 melanoma were transfected with the plasmid YFP-Globin-SL8-intron or with the PCDNA3 empty plasmid as described in the previous section. Twenty-four hours after transfection, cells were treated with different doses of isoginkgetin (Merk Millipore), IP2 or M2P2 for 18 h. Cell were then washed three times with PBS 1X and $5 \times 10^4$ cells were co-cultured with $1 \times 10^5$ SL8/H-2K$^b$-specific B3Z CD8$^+$ T-cell hybridoma for 18 h at 37 °C with 5% $CO_2$ overnight. Cells were centrifuged at 1200 rpm for 5 min, washed twice with PBS 1× and lysed for 5 min at 4 °C under shaking. Lysis buffer was composed of 0.2% TritonX-100, 10 mM DTT, 90 mM $K_2HPO_4$, 8,5 mM $KH_2PO_4$. The lysates were centrifuged at 3000 rpm for 10 min to pellet cell debris and the supernatants were transferred to a 96-well optiplate (Packard Bioscience, Randburg, SA). A revelation buffer containing 10 mM $MgCl_2$, 11,2 mM β-mercaptoethanol, 0,0015% IGEPAL® CA-630 et 40 µM 4-Methylumbelliferyl β-D-Galactopyranoside (MUG) was added and the plate was incubated at room temperature for 3 h. Finally, β-galactosidase activity was measured using the FLUOstar OPTIMA (BMG LABTECH Gmbh, Offenburg, Germany).

**FACS analysis: H-2K$^b$ expression and recovery at the cell surface.** Mouse cells were treated with drugs for 18 h. Cells were harvested and $1 \times 10^6$ cells were incubated with mouse FcR blocking reagent for 10 min at 4 °C according to the manufacturer protocol (Miltenyi Biotech). Cells were stained with anti-mouse MHC class I (H-2K$^b$)-PerCP-Cy5.5 (AF688.5, Biolegend) or with the corresponding PerCp-Cy5.5-conjugated mouse (BALB/c) IgG2a, κ isotype for 30 min at 4 °C. Cells were stained with DAPI prior to acquisition for dead cell exclusion. To study the kinetics of endogenous H-2K$^b$ recovery at the cell surface, cells were treated for 18 h with the different drugs. The day after, cells were washed and treated with ice-cold citric acid buffer (0.13 M citric acid, 0.061 M $Na_2HPO_4$, 0.15 M NaCl [pH 3]) at $1 \times 10^7$ cells per milliliter for 120 s, washed three times with PBS, and resuspended in culture medium. At the indicated time point, an aliquot of cells (generally $1.5 \times 10^6$) was collected and stained with anti-mouse H-2K$^b$ PE or anti-mouse SIINFEKL-peptide bound to H-2K$^b$-PE. All flow cytometry experiments were conducted using the BD LSRII flow cytometer (BD Biosciences) and data are analyzed with the FlowJow software (V10).

**FACS analysis: Annexin V staining.** Mouse cells were treated with drugs for 18 h. Floating and adherent cells were collected, washed in PBS and resuspended in 1× Annexin V Binding Buffer with APC Annexin V (BD Pharmingen) according to the manufacturer protocol. Cells were gently vortexed and incubated 15 min at RT in the dark. Cells were stained with DAPI prior to flow cytometry analysis. Cells were analyzed using the BD LSRII flow cytometer (BD Biosciences) and data were analyzed with the FlowJow software (V10).

**MTT assay.** Cells were plated in a 96-well plate at $1 \times 10^4$ cells per well for 24 h. Cells were then treated with increasing doses of drugs overnight. After removing the medium, the MTT (3-[4,5-Dimethylthiazol-2-yl]−2,5-diphenyltetrazolium bromide) (Sigma–Aldrich) powder was resuspended in PBS, filtered and added into the plate wells at 2.5 mg/ml. The plate was incubated for 2 h at 37 °C, with 5% $CO_2$. It was then centrifuged, and the supernatant was carefully removed in order not to disturb the precipitated formazan. Crystals were dissolved in DMSO at 200 µL/well and the plate was shaken for 10 min. Absorbance was measured at 544 nm using the FLUOstar OPTIMA.

**RNA preparation, RT, and qPCR.** Cells were transfected with the YFP-Globin-SL8-intron as described previously and then treated with drugs for 48 h. Cells were then harvested and total cellular RNA was extracted and purified using the RNeasy Mini kit (Qiagen) according to the manufacturer protocol. The reverse transcription was carried out with 0.5 µg of RNA with the iScript cDNA synthesis kit (Bio-Rad) according to the manufacturer protocol. The StepOne real-time PCR system (Applied BioSystems) was used for qPCR and the reaction was performed with the Power SYBR green PCR master mix (Applied BioSystems). Results were analyzed using the StepOne software. Mouse gene specific primers were designed as follow: 18 S mouse forward primer, 5′-GCCGCTAGAGGTGAAATTCTTG-3′; 18 S mouse reverse primer, 5′-CATTCTTGGCAAATGCTTTCG-3′; Glob-SL8-exon forward primer, 5′-AGAAGTCTGCCGTTACTGCC-3′; Glob-SL8-exon reverse primer, 5′-AGGCCATCACTAAAGGCACC-3′; Glob-SL8-intron forward primer, 5′-GTA TCAAGGTTACAAGACAG-3′; Glob-SL8-intron reverse primer, 5′-GGGAAAAT AGACCAATAGGC-3′.

**RNA-seq: read mapping, differential mRNA expression and differential splicing.** MCA205 and B16F10 cells were treated with 15 µM of isoginkgetin or 35 µM of IP2 for 18 h. RNA was extracted using QIAGEN RNeasy columns. RNA-seq libraries were prepared from total RNA by poly(A) selection with oligo(dT) beads

according to the manufacturer's instructions (Life Technologies). The resulting RNA samples were then used as input for first-strand stranded library preparation using a custom protocol based on dUTP method as described previously[1]. Libraries were sequenced on the Illumina HiSeq 4000 sequencer using 50–base pair paired-end read method. All RNA-seq data were aligned to the mouse genome assembly GRCm38 (mm10) with the TopHat2 splice-junction mapper for RNA-Seq reads (version 2.1.1) with default parameters. For the differential gene expression analysis, the Cuffdiff program (version 2.2.1)[62] was run using the following parameters: -library-type=fr-first-strand -compatible-hits-norm -library-norm-method geometric -min-reps-for-js-test 2 -dispersion-method per-condition -u -b. In all such analyses, the difference in expression of a gene was considered significant if the q value was less than 0.05 (the Cuffdiff default) and the fold change was superior to 2. Scatter plots were generated using ggplot2 in R environment. For the differential splicing analysis, the SUPPA2 program was run with the default settings[63,64].

**IP and MHC ligand elution**. $400 \times 10^6$ MCA205 or B16F10 cells were lysed in 3 mL lysis buffer containing PBS, 0.25% Sodium Deoxycholate and protease inhibitor (Pierce™ Protease Inhibitor Mini Tablets, EDTA-free, Thermo Fisher) during 1 h at 4 °C under agitation. Lysate was sonicated and then cleared by centrifugation for 1 h at $4000 \times g$ at 4 °C. Supernatant was filtered through 0.40 μm filter and applied on affinity columns overnight (Econo-Column Chromatography Columns 0.5 × 5 cm, Biorad). Flow was maintained at 10 rpm overnight by a peristaltic pomp. IP resins were prepared as follows: (i) 40 mg CNBr-activated Sepharose were activated in 1 mM HCl for 30 min at RT, (ii) 1 mg of Y-3 or B22-249 antibodies (kindly provided by Prof. Dr. Stefan Stevanović from University of Tübingen) was added in $H_2O$, 0.5 M NaCl, 0.1 M $NaHCO_3$ for 2 h at RT, (iii) resin was blocked in PBS, 0.2 M glycin for 1 h at RT, and finally (iv) resin was washed twice in PBS and resuspended at 1 mg antibody/mL in PBS. Centrifugation at 300 rpm for 4 min (no brake) and removal of the supernatant were performed between each step. On the second day, the peptides were eluted in 100 μL 0.2% trifluoroacetic acid (TFA) for 25 min at 4 °C under agitation. Elution was repeated three times. The four eluates were pooled, cleared by centrifugation for 5 min at $15,000 \times g$ and filtered through Amicon 3 kDa MWCO (Pall Nanosep® centrifugal device with Omega membrane). Prior to filtration, equilibration of the Omega membrane was performed by successive rinse/centrifugation cycles with MS-grade water, 0.1 N NaOH, MS-grade water and finally 0.2% TFA. Eventually, membrane was rinsed with 32.5% ACN, 0.2% TFA and the sample was concentrated using a Speed Vac Concentrator (Thermo Scientific). Samples were stored at −80 °C prior LC-MS/MS analysis.

**LC-MS/MS acquisition**. The peptides were desalted using ZipTip $C^{18}$ pipette tips (Pierce Thermo Scientific), eluted in 40 μL acetonitrile 70% (Fisher Scientific)/0.1% formic acid, vacuum centrifuged and resuspended in 12 μL of 0.1% formic acid. All C18 zip-tipped peptides extracts were analyzed using an Orbitrap Fusion Tribrid equipped with an EASY-Spray Nano electrospray ion source and coupled to an Easy Nano-LC Proxeon 1200 system (all devices are from Thermo Fisher Scientific, San Jose, CA). Chromatographic separation of peptides was performed with the following parameters: Acclaim PepMap100 C18 pre-column (2 cm, 75 μm i.d., 3 μm, 100 Å), Pepmap-RSLC Proxeon C18 column (75 cm, 75 μm i.d., 2 μm, 100 Å), 300 nl/min flow, gradient rising from 95% solvent A (water, 0.1% formic acid) to 28% solvent B (80% acetonitrile, 0.1% formic acid) in 105 min and then up to 40%B in 15 min followed by column regeneration for 50 min. Peptides were analyzed in the orbitrap in full ion scan mode at a resolution of 120,000 and with a mass range of $m/z$ 400–650 using quadrupole isolation and an AGC target of $1.5 \times 10^5$. Fragments were obtained by Collision Induced Dissociation (CID) activation with a collisional energy of 35%. MS/MS data were acquired in the Orbitrap at a resolution of 30,000 in a top-speed mode, with a total cycle of 3 s with an AGC target of $7 \times 10^4$. The maximum ion accumulation times were set to 100 ms for MS acquisition and 150 ms for MS/MS acquisition in parallelization mode.

**MS/MS data processing**. Raw files were converted to mzDB files[65] using a PWIZ-mzDB (https://github.com/mzdb/pwiz-mzdb) version 0.9.10, and then to MGF peaklists using a mzdb-access (https://github.com/mzdb/mzdb-access) version 0.7.0. Produced MGF files were submitted to Mascot database searches (version 2.7, MatrixScience, London, UK) against a mouse Uniprot protein sequence database released in July 2020 and downloaded from SwissProt website. The database contains 87,954 protein entries. Spectra were searched with a mass tolerance of 10 ppm in MS mode and 20 mmu in MS/MS mode. No missed cleavage was allowed (no enzyme search). Carbamidomethylation of cysteine residues and oxidation of methionine residues were set as variable modifications. For peptide validation, identification results were imported into the Proline software[66] version 2.0 (http://proline.profiproteomics.fr) for target/decoy validation and label-free quantification. Peptide Spectrum Matches (PSM) with pretty rank equal to one and a Mascot score above 20 were retained. False Discovery Rate was then optimized to be below 1% at PSM level using an adjusted variant of the Mascot E-value[67]. For label-free quantification, peptide abundances were extracted by Proline using an m/z tolerance of 5 ppm. Alignment of the LC-MS runs was performed. Cross-assignment (a.k.a. match between runs) of peptide ion abundances was performed between the raw files using an m/z tolerance of 5 ppm and a retention time tolerance of 60 s. Precursor ions abundances were normalized using the quotient normalization method[68] implemented in Proline. Fold change was calculated as the ratio of the mean normalized abundances in the IP2-treated and untreated biological replicates.

**Ex vivo assessment of peptide immunogenicity**. C57BL/6 mice were vaccinated subcutaneously at the tail base with 100 μg of synthetic long peptide mixed with 30 μG ODN 1668. One week after, $CD8^+$ T cells were isolated from spleens of vaccinated mice using magnetic selection with CD8a (Ly-2) MicroBeads (Miltenyi Biotec). $CD8^+$ T cells were co-cultured with BMDCs pulsed with 30 mg/mL of synthetic long peptides for 18 h in ELISpot plates coated with anti-IFN$_\gamma$ capture antibody (Cell Sciences). Spots were revealed according to manufacturer instructions. IL-2 concentration was measured in the supernatants of the co-culture between $CD8^+$ T cells and peptide-pulsed BMDCs using the Mouse IL-2 ELISA MAX Standard kit (BioLegend).

**In vivo experimentation: tumor challenge and drug treatment**. Female wild-type C57BL/6 mice at the age of 6 weeks were obtained from Harlan France. NU/NU nude mice were obtained from Charles River France. Animal experiments were conducted in compliance with the EU Directive 63/2010, and protocols 2016_064_5677 and were approved by the Ethical Committee of the Gustave Roussy Campus Cancer (CEEA IRCIV/IGR no. 26, registered at the French Ministry of Research). Mice were challenged subcutaneously in the right flank with $5 \times 10^4$ MCA205 wild-type (WT) or $7.5 \times 10^4$ MCA205-globin-SL8-intron cells, or with $5 \times 10^4$ B16F10 wild-type (WT) or B16F10-Globin-SL8-intron cells along with matrigel (VWR). Five days after challenge, mice were treated intraperitonealy with different dose of isoginkgetin, IP2 or M2P2. Intraperitoneal injection was repeated at day 10 and 15. Area of the tumor was recorded every 3–4 days until ethical limit points were reached. Mice which experienced complete tumor regression following treatment were kept for 100 days after tumor challenge and were then rechallenged subcutaneously on the right flank with $7.5 \times 10^4$ MCA205-Globin-SL8-intron cells and on the left flank with $5 \times 10^4$ B16F10 WT cells.

**In vivo proliferation of adoptively transferred OT-I $CD8^+$ T cells**. C57BL/6 mice were challenged subcutaneously with $3.0 \times 10^5$ MCA205-globin-SL8-intron cells and thereafter treated with 24 mg/kg of IP2 at days 8 and 11. At day 12, OT-1 $CD8^+$ T cells were isolated from spleens of OT-1 mice using magnetic selection with CD8a (Ly-2) MicroBeads (Miltenyi Biotec) and then intravenously injected into tumor-bearing mice at $3.0 \times 10^6$ cells/mouse. Three days after transfer, proliferation of OT-1 cells was measured in spleens by flow cytometry after staining with anti-CD8-FITC and APC-conjugated SL8/H-2K$^b$ Dextramers (Immudex).

**In vivo experimentation: prophylactic and therapeutic vaccination**. In the prophylactic setting, C57BL/6 mice were vaccinated twice subcutaneously at the tail base with 100 μg of synthetic long peptide mixed with 30 μG ODN 1668, two weeks apart (prime-boost). One week after the booster injection, $1.5 \times 10^5$ MCA205 WT were inoculated subcutaneously into the right flank. Mice were then treated twice a week with 24 mg/kg of IP2 for 2 weeks. In the therapeutic setting, C57BL/6 mice were injected subcutaneously in the right flank with $1.5 \times 10^5$ MCA205 WT and thereafter vaccinated once a week with 100 μg of synthetic long peptide mixed with 30 μG ODN 1668 and treated twice a week with 24 mg/kg of IP2 for 2 weeks.

**Statistics and reproducibility**. All data from T-cell activation assays were analyzed from at least three biological replicates, each one including three technical replicates. Data were presented as the means ± SEM of biological replicates. Significant changes were analyzed by a one-way analysis of variants (ANOVA) with Dunnett's post-hoc test. All in vivo experiments were repeated at least twice with a minimum of six mice per group. Significant changes between two conditions were analyzed by a Student's t-test. A one-way analysis of variants (ANOVA) with Tukey's post-hoc test was run when multiple conditions were tested. All data were processed on the GraphPad Prism (version 5) software.

**Reporting summary**. Further information on research design is available in the Nature Research Reporting Summary linked to this article.

## Data availability

The proteomics datasets are available in the PRIDE partner repository under the identification number: PXD023019 as .raw files, .dat files and Proline 2.0 label-free quantification spreadsheet at peptide level. The Source data for graphs and charts in the main figures are available from the following repository: Dryad; Source data of main figures, Dryad, Dataset https://doi.org/10.5061/dryad.0rxwdbrzb (ref. [69]). All other data that support the findings of this study are available from the corresponding author upon reasonable request.

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

## Acknowledgements

The MCA205 cell line was a gift from Prof. Laurence Zitvogel, Gustave Roussy Institute. We also thank Prof. Laurence Zitvogel for valuable comments on the manuscript. The IP antibodies were kindly provided by Prof. Dr. Stefan Stevanović from University of Tübingen. This work was funded by the FRM, project code n° DCM20181039530, The Avenir program (INSERM), La Fondation ARC, and Fondation Gustave Roussy. A.P. and R.D. were supported by the Université Paris Saclay, R.D. was also supported by the Fondation Philantropia, M.B. is supported by the Fondation Gustave Roussy.

## Author contributions

R.D. and A.P. designed and performed experiments, analyzed results, and wrote the manuscript. M.R. performed the bioinformatic analyses. D.R. synthesized and characterized the isoginkgetin derivatives. M.B. performed experiments. D.B., E.M.-B., and J.M. analyzed the MS/MS data. C.G. performed LC-MS/MS acquisition and analyzed data. M.G. provided valuable help in setting up the IP and MHC-I ligand elution protocols. M.A. supervised the synthesis and characterization of the isoginkgetin derivatives and wrote the manuscript. S.A. supervised the study, designed experiments, and wrote the manuscript.

## Competing interests

The authors declare no competing interests.
