## [Peer Review File · Communications Biology]

Reviewers' comments:

Reviewer #1 (Remarks to the Author):

Darrigrand et al. study the effect of mRNA splicing inhibitors on the generation of the MHC class I immunopeptidome. This paper is an important and powerful extension of the previous ground breaking findings of Apcher and colleagues that pre-mRNA can be an important source of antigenic peptides, particularly for tumor cells. In this unusually comprehensive soup-to-nuts study the authors characterize the spliceosome inhibitor isoginkegetin effects on peptide generation and tumor cell survival and develop novel derivatives, one of which, IP2 is both less toxic and more effective at enhancing peptide generation from pre-mRNA and demonstrates greater in vivo efficacy at activating CD8+ T cells with demonstrated anti-tumor activity in vivo. The study further demonstrates the effect of drugs the ms determined immunopeptidome, identifies novel tumor specific peptides and demonstrates their immunogenicity and effectiveness in eliciting CD8+ T cell dependent immunotherapy.

Whew! This is a ton of work, and my only comment is to applaud the authors for this protean study which has immediate translational applications.

Reviewer #2 (Remarks to the Author):

Darrigrand et al performed a very interesting study aimed to show that isoginkgetin and its phosphate water-soluble and non-toxic derivative IP2 inhibit splicosome and hence act at the stage of generating peptides from un-spliced mRNAs (pioneer translation products; PTP). In in-vitro experiments, they show that IP2 increase PTP-derived antigen presentation and in in-vivo assays they show it impairs tumor growth. They further report that IP2 changes the antigenic landscape (both coding and non-coding sequences) by analyzing the immunopeptidome of treated and untreated cells.

This is a very interesting concept and the investigation of drug-induced MHC presentation of antigens derived from retained introns is timely and challenging. However, there are concerns related to the proteo-genomics approached applied here. In addition, the manuscript would benefit from grammatical/technical/structural editorial proof-reading.

Specific concerns:

1. Material and methods section:

- Methods should be written in past tense.
- There are repetitions in several methods sections, and it is not clear if the repetitions are there for a specific reason, or just a mistake. For example, lines 277-282 and 284-292. Both sections are related to identification of peptides from MSMS data with Proteome Discoverer tool. They should be combined into one section called (for example) 'qualitative and quantitative detection of peptides'.
- Another example: lines 133-135 and 142-144. This should be fixed throughout the methods section.
- Line 138: How IP2 and M2P2 were synthesized? A detailed description should be added.
- Line 549 : "IP2 induces the presentation of tumor-specific mutated epitopes". How tumor specific

mutations were called and identified? information is missing in the methods section.

2. Technical aspects related to the application of proteogenomics and immunopeptidomics in this study should be elaborated:

- The authors mention that they have generated two databases – one called ‘retained intron’, and the second is called ‘all frame’. In the methods section, only the ‘retained intron’ database is described. The authors should add a description of the ‘all frame’ database. Were the ‘retained intron’ and ‘all frame’ references generated from RNA seq data of IP2-treated or from untreated samples?
- What is the actual size of the two databases compared with a typical proteome Uniprot reference? How did the authors validate that the size is compatible with proteogenomics application? the authors should share the generated reference fasta files with the MS data and search engine result files.
- How did the authors assess the level of error in their peptide identification process? The larger the database, the higher the chance that the best scoring match to the MSMS is incorrect, and the more difficult it becomes to distinguish between true and false identifications. In proteogenomic approaches, novel peptides identifications should require to have stronger supportive evidence than known peptides, due to the different likelihood of identifying novel vs. known peptides. When using a target decoy approach for FDR estimation, the calculation should be done separately on each class of peptides (known and novel) (PMID: 25357241). The authors should adapt a strategy dedicated for proteogenomics and add a description in the methods section to explain their FDR calculation and estimations. Was the FDR calculated separately for PSM derived from the ‘retained intron’, and the ‘all frame’ translation products (non-coding class) and the uniprot proteomes (canonical class)? Furthermore, while results obtained from different tools might be different, a good bioinformatics analysis must give consistent results even with different methods. Therefore, the authors could repeat the search analysis with another tool in order to support consistency and reproducibility of their identifications (for example PMID: 31537638).
- In line 529, the ‘all frame’ database is mentioned with a citation to a previous publication of Laumont et al. Laumont et al. generated an ‘all frame’ database, however they included only 8-11 aa long peptides and selected a combination of the Mascot score (≥ 22) and MHC-binding affinity ($\leq 1,250$ nM) for peptides identifications. In addition, they manually inspected all the MSMS identifications. These thresholds were not used in the current study. In addition, Laumont et al (2016) is a rather old reference, and more updated methods are now used for such challenging applications.
- Figure 5 A and B shows the length distribution of peptides uniquely identified in IP2-treated or untreated cells. The results of the immunopeptidomics assays are unexpected. The peptides length distribution is not typical for immunopeptidomics, and suggest a very high level of contaminants in the extracted MHC complexes samples and/or very high level of false positives. This is not only an issue of the non-coding sources (retained introns etc.), and not directly dependent on the treatment with IP2, as seen from the figure. Did the author check if the length of the identified peptides is similar when searching the data only against the canonical mouse reference?
- The authors should validate their peptide identification through independent methods (for example PMID: 32047025) and with synthetic peptides.
- The authors should report all the identified MHC peptides, not only the differentially presented ones, and provide their length distribution and assign a predicted binding affinity score to the respective HLA allotypes.

- Figure 5 E and F: the majority of the non-canonical peptides were identified in the untreated samples (marked in Blue). Also in Figure S5 C and D. The authors should explain this.
- Legends for Supplementary figures S5 and S6 are missing.

Dear Reviewers,

We would like to resubmit a revised version of our manuscript that we have now modified in order to address your different concerns. We thank both reviewers for having performed such an in-depth analysis of our study and providing extremely interesting and useful comments. In the revised manuscript, we have carefully considered each Reviewer's comments and suggestions and we have added additional explanations and figures in the letter to fully address all their concerns. All changes in the manuscript are shown in **red**. Below, all the new modifications to the manuscript carried out are described in detail in point-by-point way.

Overall, the Reviewer's comments were very helpful, and we are appreciative of such constructive feedback on our original submission. **After addressing the issues raised, we believe that the manuscript is now suitable for publication in Communications biology.**

Reviewer #1 (Remarks to the Author):

Darrigrand et al. study the effect of mRNA splicing inhibitors on the generation of the MHC class I immunopeptidome. This paper is an important and powerful extension of the previous ground breaking findings of Apcher and colleagues that pre-mRNA can be an important source of antigenic peptides, particularly for tumor cells. In this unusually comprehensive soup-to-nuts study the authors characterize the spliceosome inhibitor isoginkgetin effects on peptide generation and tumor cell survival and develop novel derivatives, one of which, IP2 is both less toxic and more effective at enhancing peptide generation from pre-mRNA and demonstrates greater in vivo efficacy at activating CD8+ T cells with demonstrated anti-tumor activity in vivo. The study further demonstrates the effect of drugs the ms determined immunopeptidome, identifies novel tumor specific peptides and demonstrates their immunogenicity and effectiveness in eliciting CD8+ T cell dependent immunotherapy. Whew! This is a ton of work, and my only comment is to applaud the authors for this protean study which has immediate translational applications.

We thank the reviewer for his very positive comments on our work and to recognize the work done by our team and especially the student Romain Darrigrand who finished his thesis in May.

Reviewer #2 (Remarks to the Author):

Darrigrand et al performed a very interesting study aimed to show that isoginkgetin and its phosphate water-soluble and non-toxic derivative IP2 inhibit splicosome and hence act at the stage of generating peptides from un-spliced mRNAs (pioneer translation products; PTP). In in-vitro

experiments, they show that IP2 increase PTP-derived antigen presentation and in in-vivo assays they show it impairs tumor growth. They further report that IP2 changes the antigenic landscape (both coding and non-coding sequences) by analyzing the immunopeptidome of treated and untreated cells.

This is a very interesting concept and the investigation of drug-induced MHC presentation of antigens derived from retained introns is timely and challenging. However, there are concerns related to the proteo-genomics approach applied here. In addition, the manuscript would benefit from grammatical/technical/structural editorial proof-reading.

Specific concerns:

1. Material and methods section:

- Methods should be written in past tense.

As required by the Reviewer, methods were modified and past tense was used throughout the section.

- There are repetitions in several methods sections, and it is not clear if the repetitions are there for a specific reason, or just a mistake. For example, lines 277-282 and 284-292. Both sections are related to identification of peptides from MSMS data with Proteome Discoverer tool. They should be combined into one section called (for example) ‘qualitative and quantitative detection of peptides’. Another example: lines 133-135 and 142-144. This should be fixed throughout the methods section.

We thank the Reviewer for these remarks and we corrected the text accordingly.

First repetition refers to the parameters used for both the qualitative and quantitative analysis of the MSMS data. The previous “LC-MS/MS acquisition and qualitative analysis” and “Quantitative analysis in label free experiments” sections were merged.

Second repetition refers to the protocol for plasmid DNA transfection. The description of the protocol was kept in the section “Cell culture, plasmid DNA transfection, drugs and peptides” and removed from the section “T-cell activation assay”.

- Line 138: How IP2 and M2P2 were synthesized? A detailed description should be added.

We thank the Reviewer for these remarks and we corrected the text accordingly. A detailed description of the synthesis of IP2 and M2P2 was added to the methods.

- Line 549 :”IP2 induces the presentation of tumor-specific mutated epitopes”. How tumor specific mutations were called and identified? information is missing in the methods section.

We thank the Reviewer for raising this point. The “retained introns” and “all frame” databases were built from the RNA-seq analysis of untreated MCA205 and B16F10 cells. Tumor specific mutations were identified after comparison with the genome reference (GENCODE release GRCm38).

2. Technical aspects related to the application of proteogenomics and immunopeptidomics in this study should be elaborated:

- The authors mention that they have generated two databases – one called ‘retained intron’, and the second is called ‘all frame’. In the methods section, only the ‘retained intron’ database is described. The authors should add a description of the ‘all frame’ database. Were the ‘retained intron’ and ‘all frame’ references generated from RNA seq data of IP2-treated or from untreated samples?

- What is the actual size of the two databases compared with a typical proteome Uniprot reference?

We thank the Reviewer for raising this point. A description of the “all frame” database was added to the methods. Both “retained intron” and “all frame” databases were generated from RNA seq data of untreated MCA205 and B16F10 cells.

The MCA205 and B16F10 “all frame” databases are composed of approximately 1.1×10^9 8-11 amino acid-long peptides (1 062 070 065 and 1 160 794 382 peptides respectively). Regarding the “retained introns” databases, we identified 135 348 retained introns for MCA205 cells and 140 943 retained introns for B16F10 cells where 124 911 were in common.

How did the authors validate that the size is compatible with proteogenomics application?

We thank the Reviewer for raising this point. We agree with the reviewer that the large size of the “all frame” databases (51 Go and 48 Go respectively) is critical in our study. For example, we were not able to perform the analysis with Mascot that could not handle that amount of data. However, the analysis was possible using Sequest.

Besides, the group of Bassani-Sternberg showed recently that database size affects false positives in nonHLAp detection (Chong et al. 2020). Nevertheless, they decided to include in their study “all non-coding transcripts with FPKM > 0 to circumvent the need to exclude polypeptide sequences based on low-expressing transcripts.” For the same reason, we believe

that for the study of non-canonical epitopes we should not exclude any sequence from the search space which implies handling huge databases.

The authors should share the generated reference fasta files with the MS data and search engine result files.

We thank the Reviewer for raising this point. Fasta files and result files should be available with PRIDE with the following information:

Project Name: Splicing inhibition enhances the antitumor immune response through increased tumor antigen presentation and altered MHC I immunopeptidome.

Project accession: PXD012102

Project DOI: Not applicable

Reviewer account details:

- *Username: reviewer38865@ebi.ac.uk*
- *Password: BdeFPkNz*

- How did the authors assess the level of error in their peptide identification process? The larger the database, the higher the chance that the best scoring match to the MSMS is incorrect, and the more difficult it becomes to distinguish between true and false identifications. In proteogenomic approaches, novel peptides identifications should require to have stronger supportive evidence than known peptides, due to the different likelihood of identifying novel vs. known peptides. When using a target decoy approach for FDR estimation, the calculation should be done separately on each class of peptides (known and novel) (PMID: 25357241). The authors should adapt a strategy dedicated for proteogenomics and add a description in the methods section to explain their FDR calculation and estimations. Was the FDR calculated separately for PSM derived from the ‘retained intron’, and the ‘all frame’ translation products (non-coding class) and the uniprot proteomes (canonical class)? Furthermore, while results obtained from different tools might be different, a good bioinformatics analysis must give consistent results even with different methods. Therefore, the authors could repeat the search analysis with another tool in order to support consistency and reproducibility of their identifications (for example PMID: 31537638).

We thank the Reviewer for raising this point. The “Fixed Value PSM Validator node” were used to validate PSMs and peptides without performing decoy search but based on fixed score thresholds (XCorr confidence thresholds) defined in the Sequest HT search engine.

As suggested by the reviewers we tried to repeat the search analysis with another tool. During these 3 months, we tried to find a bioinformatician and an MS/MS platform to re analyze our

data, as suggested by reviewer n°2. Unfortunately, we could not find in France or in any other European country an MS/MS platform to re-analyze our data for many reasons. The first is related to the current health situation. Indeed, many mass spectrometry platforms had to work with a limited number of people. As a result, all the platforms we contacted, as well as the IJM platform that had already analyzed our samples, responded negatively to our requests. The second reason is related to the massive size of our databases (59,065,766,536 bytes). Most platforms did not want to take the risk/time to download such huge files to their server, which would have taken, for some platforms, more than a month. Eventually, our databases cannot be used with the MASCOTT software but only with the Sequest software. Some of the platforms contacted do not use this software. Therefore, we could not find a platform to work with such databases. We were waiting for the last answer last week from a platform in Toulouse and they declined our offer too.

Nevertheless, we were able to validate the presence of peptide MCA205 KB-1 TNQDFIQRL with FDR5% with a new pipeline (Mascot + Proline). This is an important result since this peptide was shown to be immunogenic in vivo. The annotated spectrum of peptide TNQDFIQRL is shown below. A score of 51 was calculated for this peptide on Proline.

- In line 529, the ‘all frame’ database is mentioned with a citation to a previous publication of Laumont et al. Laumont et al. generated an ‘all frame’ database, however they included only 8-11 aa long peptides and selected a combination of the Mascot score (≥ 22) and MHC-binding affinity ($\leq 1,250$ nM) for peptide identifications. In addition, they manually inspected

all the MSMS identifications. These thresholds were not used in the current study. In addition, Laumont et al (2016) is a rather old reference, and more updated methods are now used for such challenging applications.

We thank the Reviewer for these remarks. For the generation of our “all frame” database we took advantage of the methods firstly described by Laumont et al. in 2016 and built, from RNA-seq data, a non-canonical cancer database encompassing the peptides generated from all the regions of the genome. It is true that we did not apply the same filters such as Mascot Score and MHC-binding affinity for peptide selection. We used the correlation score (XCorr) on Sequest since the search could not be performed with Mascot. The predictive binding affinity was calculated with NetMHC but was not a selection criterion for the peptides. The epitope B16F10 DB-2 does not bind to Db molecules according to the NetMHC algorithm while we found it at the cell surface and it was shown to be immunogenic both ex vivo and in vivo. This interesting peptide would have been missed if the predictive binding affinity was a selection criterion in our study.

- Figure 5 A and B shows the length distribution of peptides uniquely identified in IP2-treated or untreated cells. The results of the immunopeptidomics assays are unexpected. The peptides length distribution is not typical for immunopeptidomics, and suggest a very high level of contaminants in the extracted MHC complexes samples and/or very high level of false positives. This is not only an issue of the non-coding sources (retained introns etc.), and not directly dependent on the treatment with IP2, as seen from the figure. Did the author check if the length of the identified peptides is similar when searching the data only against the canonical mouse reference?

As requested by the reviewer, we compared the length distribution of the peptides when identified on the “Uniprot extracted” database vs “Retained Introns” database vs “Uniprot+Retained Introns” concatenated database. We observed that the peptides identified on the reference Uniprot extracted database followed a typical length distribution for H-2Kb and H-2Db epitopes with a preference for 8mers and 9mers, respectively (Stevens et al. 1998). However, retained intron-derived epitopes appeared to be longer. As retained intron-derived epitopes represent 90% of the peptides identified on the “Retained Introns + Uniprot” concatenated database, the length distribution is mostly dictated by the longer retained intron-derived epitopes.

Besides, around 50% of all the H-2Kb epitopes and 75% of the 8mers identified on the reference Uniprot extracted database are predicted binders according to NetMHC 4.0 algorithm. Similarly, around 40% of all the H-2Db epitopes and 65% of the 9mers identified on the reference Uniprot extracted database are predicted binders according to NetMHC 4.0 algorithm. The coherence of the results obtained for the canonical epitopes identified on Uniprot strengthen our confidence in the results obtained for the non-canonical epitopes.

- The authors should validate their peptide identification through independent methods (for example PMID: 32047025) and with synthetic peptides.

We agree with the reviewer that validation of peptide identification with synthetic peptides is a strong asset that we contemplate to use in our coming studies. However, in the context of this article, we do not have the technical and human resources to address this issue anymore.

- The authors should report all the identified MHC peptides, not only the differentially presented ones, and provide their length distribution and assign a predicted binding affinity score to the respective HLA allotypes.

We thank the Reviewer for these remarks. We reported in two attached excel files the list of all identified MHC peptides in MCA205 and B16F10 samples. Peptides were sorted by confidence of identification (HIGH or MEDIUM) and length. For each peptide a predicted binding affinity score was calculated on NetMHC. Predicted strong binders ($IC_{50} < 50nM$)

are highlighted in green, predicted weak binders ($IC_{50} < 500nm$) are highlighted in red. The length distribution of the peptides is also provided for each database in the different excel files.

- Figure 5 E and F: the majority of the non-canonical peptides were identified in the untreated samples (marked in Blue). Also in Figure S5 C and D. The authors should explain this.

We thank the Reviewer for raising this interesting point. The peptides displayed in figures 5E and 5F are shared between untreated and treated cells but their presentation at the cell surface is significantly altered upon treatment. The peptides marked in blue are significantly less present at the surface of MCA205 or B16F10 cells upon treatment. In B16F0 cells, it appears that IP2 treatment mostly reduce the presentation of some MHC-I epitopes while a few are significantly upregulated. We have shown that some peptides upregulated upon treatment with IP2 are immunogenic, it could be interesting to assess the immunogenicity of the downregulated peptides to understand if their specific loss contribute to the enhanced immunogenicity of the cancer cells. This experiment is actually done in my laboratory for the following story where we aim to show that the downregulation of the expression of some peptides is also beneficial for the immune system to attack tumors. In fact our preliminary results demonstrate that some peptides that are downregulated were capable to bind strongly to the MHC class I molecule without any immunogenicity behind this binding.

- Legends for Supplementary figures S5 and S6 are missing.

We thank the Reviewer for these remarks and we corrected the text accordingly. Legends for supplementary figures S5 and S6 have been added to the manuscript.

Legends:

Chong, Chloe, Markus Müller, HuiSong Pak, Dermot Harnett, Florian Huber, Delphine Grun, Marion Leleu, et al. 2020. « Integrated Proteogenomic Deep Sequencing and Analytics Accurately Identify Non-Canonical Peptides in Tumor Immunopeptidomes ». Nature Communications 11 (1): 1293. <https://doi.org/10.1038/s41467-020-14968-9>.

Stevens, James, Karl-Heinz Wiesmüller, Peter Walden, et Etienne Joly. 1998. « Peptide Length Preferences for Rat and Mouse MHC Class I Molecules Using Random Peptide Libraries ». European Journal of Immunology 28 (4): 1272-79. [https://doi.org/10.1002/\(SICI\)1521-4141\(199804\)28:04<1272::AID-IMMU1272>3.0.CO;2-E](https://doi.org/10.1002/(SICI)1521-4141(199804)28:04<1272::AID-IMMU1272>3.0.CO;2-E).

Reviewers' comments:

Reviewer #1 (Remarks to the Author):

Authors have done an excellent job responding to comments and suggestions. Fine study!

Reviewer #2 (Remarks to the Author):

Please check the attached file.

Dear authors,

It is well accepted that proteogenomic-based MS searches of huge databases must be done carefully, with dedicated bioinformatics pipelines and solutions to avoid the propagation of false positive. The challenges and the prevalence of false identifications in immunopeptidomics studies resulting from the application of poor bioinformatics and lack of experimental validation have been the focus of a debate in recent years, see examples in following references:

Liepe, J. et al. A large fraction of HLA class I ligands are proteasome-generated spliced peptides. *Science* 354, 354-358, doi:10.1126/science.aaf4384 (2016).

Mylonas, R. et al. Estimating the Contribution of Proteasomal Spliced Peptides to the HLA-I Ligandome. *Molecular & cellular proteomics : MCP* 17, 2347-2357, doi:10.1074/mcp.RA118.000877 (2018).

Rolfs, Z., Solntsev, S. K., Shortreed, M. R., Frey, B. L. & Smith, L. M. Global Identification of Post-Translationally Spliced Peptides with Neo-Fusion. *Journal of proteome research*, doi:10.1021/acs.jproteome.8b00651 (2018).

Erhard, F., Dölken, L., Schilling, B. & Schlosser, A. Identification of the cryptic HLA-I immunopeptidome. *canimm.0886.2019*, doi:10.1158/2326-6066.CIR-19-0886 %J Cancer Immunology Research (2020)

I still have major concerns about the validity of the immunopeptidomics results, especially after performing a short simple analysis (provided below) of the list of identified peptides that the authors provided with the revised manuscript. It is not so surprising that proteomics core facilities have refused to re-analyze the data, as MS searches of huge databases for immunopeptidomics studies are highly challenging, they require advanced bioinformatics pipelines, and expert should do them in order to overcome the inherent probability of propagating false positives.

Because the database generated in this study was too large, the authors didn't use a decoy search approach to estimate the error level. Instead, the Sequest "Fixed Value PSM Validator node" was used to retain peptide spectrum matches. In the revised version the authors indicated that "XCorr threshold for high confidence was set to 1.5 for z=1, 2 for z=2, 2.5 for z=3 and 3 for z ≥ 4; XCorr threshold for medium confidence was set to 0.7 for z=1, 0.9 for z=2, 1.2 for z=3 and 1.5 for z ≥ 4. In a final consensus workflow, only peptides with at least medium confidences are considered for MCA205 and B16F10 samples." The above thresholds for 'high' confidence identifications are probably fine for some very basic shotgun proteomics analyses, but they are not suitable for immunopeptidomics ('no-enzyme search'), and the thresholds for medium confidence are absolutely too low.

The peptide length distribution analysis that was provided by the authors in the rebuttal is reassuring that the MHC-peptide samples are overall of good quality, somehow of low coverage (hundreds of peptides only) but still enriched with MHC peptides, as can be seen in the 'uniprot' search – mainly 8 or 9 mers as expected. The different length distribution obtained when the MS data was matched against the large databases was not explained by the authors. It highlights major issues with MHC-peptide identifications. The peptides identified with 'high' confidence have the expected length distribution and the peptides largely fit the binding motifs, though they constitute only a small fraction of all identified peptides. Most of the peptides are those identified with 'medium' thresholds.

Here, the peptides per length (8, 9 or 10 mers) identified in the MCA205 sample in the 'Uniprot' search were clustered to reveal the binding motifs:

With the 'high' score thresholds used, almost 98% of the peptides were included in the cluster, suggesting overall a high fraction of MHC-binders. Although the level of error cannot be estimated with this simple analysis, the results are reasonable.

With the 'medium' score, a drastic drop in the fraction of peptides that fit the binding motifs is observed:

With the 'medium' threshold, between 18-41% of the peptides were clustered in a trash cluster. This is in agreement with the very low fraction of peptides that are predicted to bind the respective MHC molecules, as provided by the authors in the new datasets provided. The conclusion here is that the 'medium' threshold is not appropriate even in case of a simple search against standard database as Uniprot.

The same analysis was done for the 'retained intron+Uniprot' database as an example. With 'high' confidence threshold:

More than 82% of the peptides were clustered (the remaining were in the trash cluster), and the motifs are similar to those observed with the 'Uniprot' only search, but less specific, suggesting some level of error.

With the 'medium' threshold:

No binding motifs were observed with this data suggesting that almost all the identified peptides here are not MHC ligands and therefore the peptides were wrongly identified. This observation is in agreement with the MHC binding prediction scores that were provided by the authors in the new supplementary excel tables. The results related to the B16 cell line and for the larger 'all frames' database are expected to follow the same trend.

The 'medium' thresholds applied by the authors for peptide identifications that led to the selection of peptides for differential presentation analysis and to the selection of peptides for the immunization assays are not acceptable. No conclusion can be made on the magnitude of and the contribution of PTPs to the MHC ligandome. The entire section on immunopeptidomics in this manuscript is not suitable for publication in its current state.

Important to note that this critics is not related to the in vitro or in vivo activity of the inhibitors, or to the fact that immunization with some intronic regions is advantageous in combination with the inhibitors. I support the publication of the manuscript without the immunopeptidomics section following revisions to the text throughout to correct for this change. However, many of the peptides identified as 'medium' confidence peptides were used for the immunization assays. The authors would need to justify their selection without MS data (random selection?). Furthermore, most of the selected peptides were also those predicted as non-binders. The authors should explain the presentation of peptides with such low binding affinity, while the Uniprot-derived peptides were of high affinity. Why the authors selected such low affinity peptides for this set of experiments? Could there be potentially other peptides included in the long peptides used in the vaccination that could lead to an immune reponse?

If the authors insist to include the MS part (entirely or only the 'high' confidence identifications), they would need to justify the selection of the confidence score thresholds, to provide computational or experimental validation and extensive explanations to support their data and conclusions: Information about the size of the 'all frame' database should be included in the text. "The MCA205 and B16F10 "all frame" databases are composed of approximatively 1.1×10^9 8-11 amino acid-long peptides (1 062 070 065 and 1 160 794 382 peptides respectively)". The excel tables (called dataset for reviewer) should be included as supplementary tables. The length distribution provided in the rebuttal should be included in the manuscript and explained. Discussion should be added on the mechanisms of how longer peptides could be generated particularly from the non-canonical and intronic sources, as well as discussion on the difference in peptide length upon treatment when the drug is affecting RNA splicing and not translation or peptide processing. Explanation on why peptides predicted as non-binders were selected for the immunization should be included discussed. Could there be other peptides included within the long peptides that could lead to an immune reponse upon immunization? Some of the minimal epitopes (in red) used in the pools (Sup Table 5) were not found in the 'datasets for reviewer'.

Dear Editor,

In the previous reviewing process, Reviewer 2 expressed concerns about the validity of the immunopeptidomic results presented in our study but supported the publication of the manuscript without the immunopeptidomic section.

We have paid much attention to the reviewer's comments and suggestions and we propose here a new version of our paper where the immunopeptidomic section was largely and significantly modified and the introduction, the results and the discussion were revised accordingly.

As suggested by the reviewer 2, we have decided to exclude the data obtained on the huge "All frame" and "Retained Intron" databases. We therefore focus the study on the modifications induced by IP2 treatment on the canonical MHC-I immunopeptidome of MCA205 fibrosarcoma. The analysis was carried out by Julien Marcoux, David Bouyssié and Emmanuelle Mouton from the "Proteomics and Mass Spectrometry of Biomolecules" team at the Institute of Pharmacology and Structural Biology (IPBS) in Toulouse. The three scientists co-sign the manuscript we propose for publication today.

The mass spectrometry proteomics data have been deposited to the ProteomeXchange Consortium via the PRIDE [1] partner repository with the dataset identifier PXD023019.

Reviewer account details:

Username: reviewer_pxd023019@ebi.ac.uk

Password: zZAbOSRz

Because we focused on the canonical MHC-I immunopeptidome, the search was performed on the reference murine proteome available on Uniprot. As detailed in the Material and Methods section, target/decoy validation and label-free quantification were performed on Proline. We share here the peptides identified with an optimized False Discovery Rate below 1% in the different replicates of untreated and IP2-treated MCA205 fibrosarcoma.

The validated peptides follow a conventional length distribution for MHC-I epitopes which is now provided in Fig 5A and copied below :

The peptides identified were clustered to reveal the binding motifs:

H-2KB 8mers

H-2KB 9mers

H-2DB 9mers

H-2DB 10mers

The length distribution, the predicted binding affinity and the clustering support the validity of the identified MHC-I epitopes and the pertinence of our analysis of the alterations of the MHC-I immunopeptidome upon IP2 treatment.

REVIEWERS' COMMENTS:

Reviewer #2 (Remarks to the Author):

The current version of the manuscript is much improved. Only a few minor comments should be addressed before the manuscript would be ready to be published:

Line 677: the authors mention “neoepitopes that are produced only upon treatment with IP2”. This is an overstatement, as based on this single experiments one cannot conclude that these are really only produced upon IP2 treatment, and not by other mechanisms. This statement should be toned down.

Line 679: “is undoubtedly in keeping with the present study”, is not clear and should be written differently.

Line 698: regarding the D>A switch. The > sign is typically used for nucleotide change and not for amino acids, and the direction is wrong. According to the sequence provided in figure S6, the switch is from Alanine to Aspartic acid. This should be corrected.

Line 699-702: it is not clear what the authors discuss here.

The text implies as if the ‘endogenous’ is the actual presentation in the cells and the mutation is only relevant in the SLP, but this is not the case. The word ‘introduced’ is also misleading in this context, because the mutation was detected at the RNAseq and was not introduced artificially to enhance presentation of the SLP. It is also not clear what the authors mean with ‘endogenous’ nischarin - is it epitope produce from the non-mutated gene (non MCA205 cells) or is this here to distinguish between the SLP generated epitope in the vaccination experiments and the naturally presented epitopes in MCA205 cells? or in non-cancerous cells?

Perhaps the sentence should be written: ...we hypothesize that this mutation found in the nischarin gene in MCA205 cells could potentially play a role in the increase presentation of the epitope compared with other healthy tissues. Future work will be needed to explore the effect of IP2 treatment on the presentation of the TL9 epitopes with and without this mutation, as well as other IP2-induced MHC ligands in cancerous and non-cancerous cells.

Dear editor, dear reviewers

We are pleased to present you this last version of our manuscript where we addressed the remaining concerns of the reviewers.

The modifications will appear in red in the manuscript. We provide below a point-by-point answer to the reviewer's comments and suggestions.

Reviewer #2 (Remarks to the Author):

The current version of the manuscript is much improved. Only a few minor comments should be addressed before the manuscript would be ready to be published:

Line 677: the authors mention "neoepitopes that are produced only upon treatment with IP2". This is an overstatement, as based on this single experiments one cannot conclude that these are really only produced upon IP2 treatment, and not by other mechanisms. This statement should be toned down.

We cannot indeed exclude that other mechanisms independent from IP2 treatment can lead to the production of those epitopes in different biological contexts. Therefore, we now refer to those epitopes absent from untreated cells as "epitopes that appear upon treatment".

Line 679: "is undoubtedly in keeping with the present study", is not clear and should be written differently.

We now state that the study of the plasticity of non-conventional antigens upon treatment with IP2 "is worth further investigation".

Line 698: regarding the D>A switch. The > sign is typically used for nucleotide change and not for amino acids, and the direction is wrong. According to the sequence provided in figure S6, the switch is from Alanine to Aspartic acid. This should be corrected.

We confirm that the switch is from Alanine (from the non-mutated gene) to Aspartic acid (from the mutated version identified in the RNAseq. The use of the sign > was inappropriate.

Line 699-702: it is not clear what the authors discuss here. The text implies as if the 'endogenous' is the actual presentation in the cells and the mutation is only relevant in the SLP, but this is not the case. The word 'introduced' is also misleading in this context, because the mutation was detected at the RNAseq and was not introduced artificially to enhance presentation of the SLP. It is also not clear what the authors mean with 'endogenous' nischarin - is it epitope produce from the non-mutated gene (non MCA205 cells) or is this here to distinguish between the SLP generated epitope in the vaccination experiments and the naturally presented epitopes in MCA205 cells? or in non-cancerous cells?

Perhaps the sentence should be written: ...we hypothesize that this mutation found in the nischarin gene in MCA205 cells could potentially play a role in the increase presentation of the epitope compared with other healthy tissues. Future work will be needed to explore the effect of IP2 treatment on the presentation of the TL9 epitopes with and without this mutation, as well as other IP2-induced MHC ligands in cancerous and non-cancerous cells.

We confirm that the mutation was detected at the RNAseq and not introduced artificially. By “endogenous” nischarin we meant the sequence in amino acid of the nischarin protein translated from the non-mutated gene that can be found in the mouse genome assembly GRCm38 (mm10). The mutation was found in the nischarin gene in MCA205 fibrosarcoma independently of IP2 treatment, but the epitope TNQDFIQRL (TL9) was found enriched upon IP2 treatment. However, the mutation is not within the TL9 epitope itself but within the expected flanking regions of this epitope that we used to build the TL9 SLP.

We agree with the modified version of this paragraph and thank the reviewer for his proposition.